# A conformational sensor based on genetic code expansion reveals an autocatalytic component in EGFR activation

Martin Baumdick[1], Márton Gelléri[1,2], Chayasith Uttamapinant[3], Václav Beránek[3], Jason W. Chin [3] & Philippe I.H. Bastiaens[1,2]

Epidermal growth factor receptor (EGFR) activation by growth factors (GFs) relies on dimerization and allosteric activation of its intrinsic kinase activity, resulting in trans-phosphorylation of tyrosines on its C-terminal tail. While structural and biochemical studies identified this EGF-induced allosteric activation, imaging collective EGFR activation in cells and molecular dynamics simulations pointed at additional catalytic EGFR activation mechanisms. To gain more insight into EGFR activation mechanisms in living cells, we develop a Förster resonance energy transfer (FRET)-based conformational EGFR indicator (CONEGI) using genetic code expansion that reports on conformational transitions in the EGFR activation loop. Comparing conformational transitions, self-association and auto-phosphorylation of CONEGI and its Y845F mutant reveals that $Y_{845}$ phosphorylation induces a catalytically active conformation in EGFR monomers. This conformational transition depends on EGFR kinase activity and auto-phosphorylation on its C-terminal tail, generating a looped causality that leads to autocatalytic amplification of EGFR phosphorylation at low EGF dose.

---

[1] Department of Systemic Cell Biology, Max Planck Institute of Molecular Physiology, Otto-Hahn-Street 11, 44227 Dortmund, Germany. [2] Faculty of Chemistry and Chemical Biology, Technical University Dortmund, Otto-Hahn-Street 6, 44227 Dortmund, Germany. [3] Medical Research Council Laboratory of Molecular Biology, Cambridge Biomedical Campus, Francis Crick Avenue, Cambridge CB2 0QH, UK. Correspondence and requests for materials should be addressed to J.W.C. (email: chin@mrc-lmb.cam.ac.uk) or to P.I.H.B. (email: philippe.bastiaens@mpi-dortmund.mpg.de)

Dimerization of EGFR by GFs activates its intrinsic kinase activity, which trans-phosphorylates tyrosine residues on the C-terminal receptor tail[1,2]. SH2- or PTB-containing signal transducing proteins are then recruited to these phosphorylated tyrosines, propagating the signal in the cytoplasm[3,4]. Structural data of EGFR indicate that in absence of ligand, a closed tethered extracellular domain (ECD) and association of the intracellular tyrosine kinase domain (TKD) with the negatively charged plasma membrane (PM) by two polybasic stretches favor steric auto-inhibition of EGFR's intrinsic kinase activity[5-7]. Ligand binding to EGFR is coupled to conformational changes in the extra- and intracellular domains and overcomes intrinsic auto-inhibition resulting in allosteric activation via asymmetric dimer formation of the TKD[2,5]. For this, the αC-helix located in the N-lobe of the TKD moves from its out-configuration to an ordered in-configuration, while the activation loop frees the catalytic cleft and undergoes conformational rearrangements of ~20 Å[8-10]. Despite the steric auto-inhibitory features, autonomous EGFR phosphorylation was observed in several cancer types including breast and lung cancer that either exhibit high EGFR surface concentrations through EGFR overexpression or bear oncogenic mutations favoring an active conformation[9,11-13]. Spontaneous auto-phosphorylation of unliganded EGFR can occur due to thermal fluctuations that overcome intrinsic steric auto-inhibition[14-16]. These auto-phosphorylation events can trigger an autocatalytic amplification mechanism when they induce an active conformation that further catalyzes EGFR auto-phosphorylation[15]. Molecular dynamics simulations suggested that $Y_{845}$ phosphorylation in the EGFR activation loop suppresses intrinsic disorder in the αC-helix, thereby stabilizing an active receptor conformation as well as increasing EGFR dimerization[9]. We therefore investigate whether EGFR can adopt an active conformation upon $Y_{845}$ phosphorylation and how this impacts on collective EGFR phosphorylation dynamics in living cells. A clear avenue to obtain a better insight in collective EGFR activation is to monitor conformational dynamics of the TKD. For this, we engineer a FRET-based conformational EGFR indicator (CONEGI) using genetic code expansion. In contrast to existing kinase activity sensors based on substrate phosphorylation[17], CONEGI is designed to report on conformational transitions in a functional domain of the EGFR TKD by the change in distance and orientation of a fluorophore conjugated to an unnatural amino acid (UAA) relative to the fluorescent protein mCitrine inserted into a conformationally invariant region. Based on structural data, we identify the end of the TKD as an insertion site for mCitrine that is conformationally invariant and does not affect EGFR function. UAA incorporation and subsequent site-specific labeling at position 851 creates a FRET-based sensor that reports on conformational transitions of the EGFR activation loop. This construct retains EGFR dimerizing and catalytic functionality. Monitoring conformational transitions together with dimerization and auto-phosphorylation and comparing these readouts to a CONEGI Y845F mutant reveals that an active conformation in monomeric receptors is induced by $Y_{845}$ phosphorylation. We then show that $Y_{845}$ phosphorylation depends on auto-phosphorylation of the C-terminal tail, which creates an autocatalytic loop that amplifies EGFR phosphorylation at low, non-saturating EGF concentrations.

## Results

**Design and performance of CONEGI.** To monitor conformational states of the TKD, we engineered multiple FRET-based conformational EGFR sensor variants, in which the donor, monomeric Citrine (mCitrine), was always genetically encoded at the same rigid region of the TKD and genetic code expansion and bioorthogonal labeling chemistry were used to position the acceptor, Atto590, at different, flexible structures of the TKD that change conformation upon activation (Fig. 1a). This hybrid sensor design allows measuring structural changes in different key functional TKD regions relative to the donor. Conformational movements of these TKD regions will alter the distance and angle between mCitrine and Atto590 resulting in changes in FRET efficiency, which can be quantified by fluorescence lifetime imaging microscopy (FLIM). mCitrine was selected as donor because of its mono-exponential fluorescence decay profile[18,19] and the membrane-permeable Atto590 as acceptor because of its high extinction coefficient ($\varepsilon$: 120,000), high quantum yield (QY: 0.8), and spectral overlap with mCitrine ($R_0 = 5.9$ nm).

By aligning active (red; PDB: 2J5F) and inactive (cyan; PDB: 2GS7) crystal structures of the EGFR TKD (Fig. 1b), we identified a conformationally invariant region at the TKD end, where we inserted mCitrine between amino acids (aa) $Q_{958}$ and $G_{959}$ (EGFR-QG-mCitrine)[15]. This site is exposed to the protein surface and not part of the asymmetric as well as the symmetric dimerization interface or regions known to be essential for kinase activity (Supplementary Fig. 1a). To further minimize perturbation of the TKD structure and constrain the mCitrine orientation, we used two linkers that form an antiparallel coiled-coil helix for mCitrine insertion[20]. EGFR-QG-mCitrine was fully active and correctly localized as apparent from its similar localization and EGF-induced phosphorylation as compared to C-terminally tagged EGFR (EGFR-mCitrine) (Fig. 1c; Supplementary Fig. 1b, c), which was shown to follow the localization and activity of endogenous EGFR[21,22]. We then selected three regions (kinase loop 2, αC-helix, and activation loop) that showed substantial differences between the active and inactive conformation for Atto590 attachment (Fig. 1b). We replaced the coding sequences of amino acids $K_{713}$ in the kinase loop 2, $K_{730}$ and $D_{737}$ in the αC-helix and $K_{843}$ and $K_{851}$ in the activation loop with an amber codon and also inserted an amber codon between the coding sequence for $E_{712}$ and $K_{713}$. This resulted in the creation of a series of EGFR(TAGXXX)-QG-mCitrine variants, where XXX indicates the amber codon position. We estimated the distances between the QG site that included the maximum linker length and each proposed site of Atto590 attachment. The obtained distances ($R$) of ~7.4–8.5 nm were sufficiently close to the Förster radius ($R_0 = 5.9$ nm) of the employed FRET pair resulting in estimated FRET efficiencies ($E_{FRET}$) between 10 and 21% for the different variants as calculated by $E_{FRET} = \frac{R_0^6}{R_0^6 + R^6}$ (Supplementary Table 1). This showed that the 851 site yielded the highest (21%) FRET efficiency. We conjugated Atto590 to the desired site in the TKD in a two-step process to generate conformational EGFR indicators (CONEGIs): first, we used a derivative of the pyrrolylsyl-tRNA synthetase/tRNA_CUA pair from *Methanosarcina mazei* to express each EGFR(TAGXXX)-QG-mCitrine gene and direct the incorporation of added UAA bicyclo[6.1.0]nonyne-lysine (BCNK)[23,24]; second, we labeled each EGFR(BCNKXXX)-QG-mCitrine variant, where XXX indicates the position of BCNK incorporation, with a tetrazine-Atto590 (tet-Atto590) conjugate via an inverse electron demand Diels–Alder reaction.

Expression of EGFR(BCNKXXX)-QG-mCitrine variants in HEK293T cells was dependent on BCNK (Fig. 1d). Whereas EGFR(BCNK851)-QG-mCitrine was expressed at a comparable level to EGFR-QG-mCitrine, all other variants exhibited reduced expression (Fig. 1d). EGFR(BCNKXXX)-QG-mCitrine labeling with tet-Atto590 was selective, as judged by both, fluorescence imaging of cell lysates following SDS-PAGE, and co-localization of mCitrine and Atto590 fluorescence at the PM and intracellular compartments as visualized in living cells by confocal microscopy (Fig. 1d, e). All EGFR(BCNKXXX)-QG-mCitrine variants

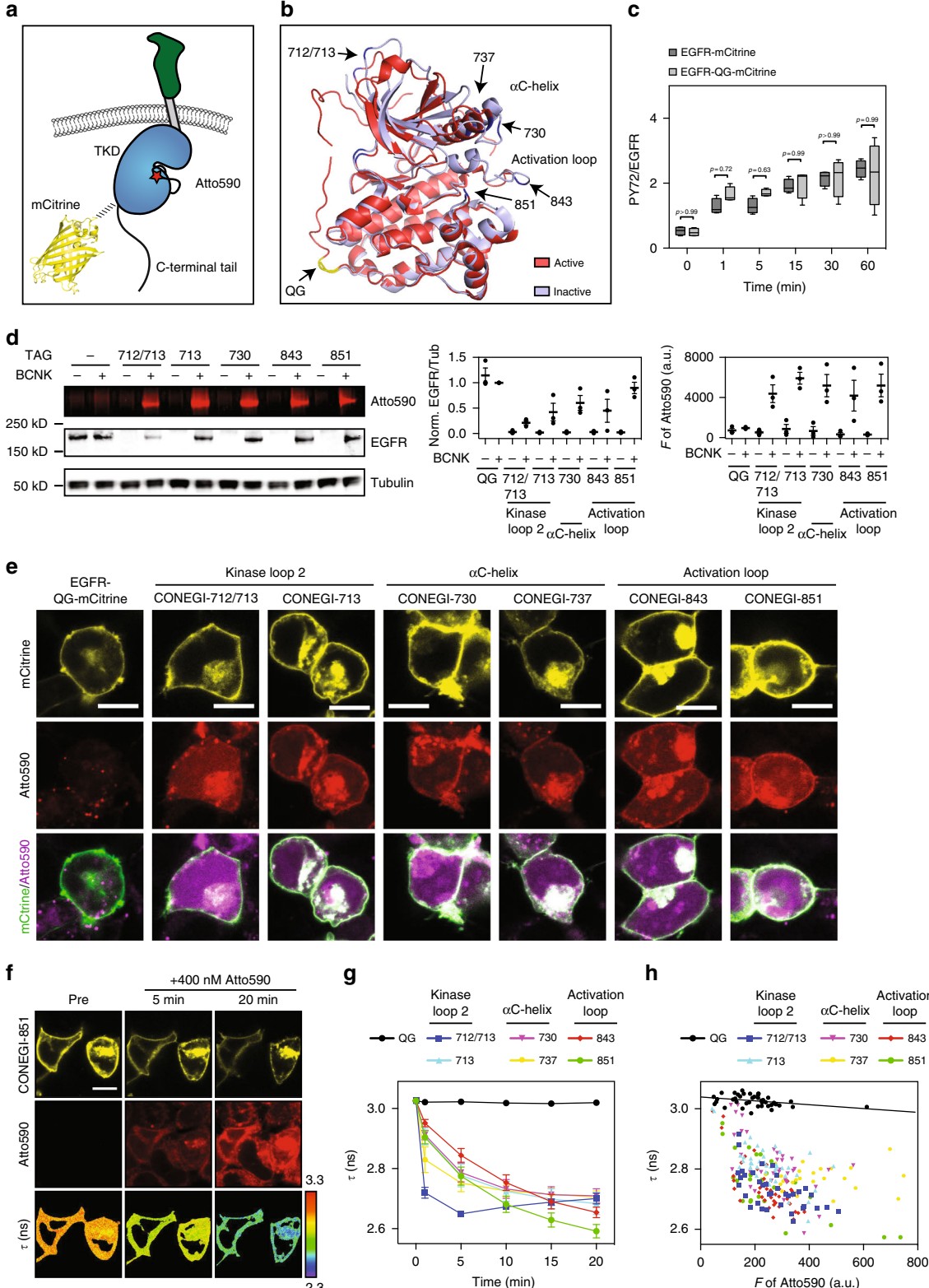

exhibited a similar localization as compared to an EGFR variant C-terminally tagged with mTurquoise (EGFR-mTurquoise) (Supplementary Fig. 1d). In addition to their comparable PM distribution (Supplementary Fig. 1d, e), all CONEGI constructs exhibited pericentriolar localization similar to EGFR-mTurquoise. This pericentriolar compartment was identified to

be the Rab11-positive recycling endosome (Supplementary Fig. 1f), which was previously shown to maintain EGFR at the PM by continuous recycling[15,25].

To experimentally determine whether EGFR(BCNKXXX)-QG-mCitrine labeling with Atto-590 results in FRET, we imaged the fluorescence lifetime ($\tau$) of mCitrine at the PM after addition of

**Fig. 1** Design and performance of CONEGI. **a** Schematic representation of CONEGI. mCitrine is fused to the C-terminal end of the tyrosine kinase domain (TKD) using a coiled-coil linker (dashed line) and Atto590 (red star) is site-specifically attached to the activation loop. **b** Alignment of active (red; PDB: 2J5F) and inactive (cyan; PDB: 2GS7) crystal structures of the EGFR TKD. mCitrine insertion (QG, yellow) and BCNK incorporation sites (blue, black arrows) are indicated. **c** Relative phosphorylation (PY72/EGFR) of EGFR-mCitrine or EGFR-QG-mCitrine upon EGF stimulation determined by western blot analysis ($n = 4$; two-way analysis of variance (two-way ANOVA)) (Supplementary Fig. 1b). **d** Fluorescence images and western blot analysis following SDS-PAGE of HEK293T cell lysates showing Atto590 fluorescence and expression level of EGFR-QG-mCitrine and CONEGIs depending on BCNK. Blots were probed with anti-EGFR and anti-Tubulin (left). Normalized relative EGFR expression (EGFR/Tub) (middle) and Atto590 fluorescence intensity (right) of EGFR-QG-mCitrine and CONEGIs ($n = 3$ blots). **e** Representative mCitrine and Atto590 fluorescence images of EGFR-QG-mCitrine and CONEGIs in HEK293T cells and corresponding green/magenta overlay. **f** Representative fluorescence images of EGFR(BCNK851)-QG-mCitrine upon tetrazine-Atto590 labeling and corresponding $\tau$. **g** Mean $\tau$ in CONEGIs and EGFR-QG-mCitrine (QG: $n = 10$ cells; 712/713: $n = 8$; 713: $n = 6$; 730: $n = 11$; 737: $n = 7$; 843: $n = 10$; 851: $n = 17$) at the PM upon tetrazine-Atto590 addition. **h** Dependency of $\tau$ in CONEGIs and EGFR-QG-mCitrine (QG: $n = 42$ cells; 712/713: $n = 36$; 713: $n = 43$; 730: $n = 36$; 737: $n = 30$; 843: $n = 44$; 851: $n = 32$) on mean Atto590 fluorescence intensity ($F$ of Atto590) at the PM after 20 min labeling. Data points represent individual cells. $\tau$, fluorescence lifetime of mCitrine. Scale bars, 10 μm. EGF stimulation, 100 ng ml$^{-1}$. Error bars: SEM, except Fig. 1c: Tukey box plot

tet-Atto590 to living cells by FLIM. For all CONEGIs, we obtained a significant decrease in $\tau$ of mCitrine from $3.02 \pm 0.004$ ns (SEM) to $2.67 \pm 0.039$ ns (SEM) over a 20 min time course, corresponding to an average $E_{FRET}$ between 10 and 14% as calculated by: $E_{FRET} = 1 - \frac{\tau_{DA}}{\tau_D}$, where Atto590 conjugation to BCNK incorporated into the 851 site yielded the highest FRET efficiency ($14.3 \pm 0.8\%$ (SEM)) (Fig. 1f, g). These experimental $E_{FRET}$ values were in broad agreement with the theoretical predictions, exhibiting equal or lower values possibly due to the relative orientation between the dyes, which affects the parameter $\kappa^2$ in $R_0$ (Supplementary Table 1, Fig. 1g). CONEGI-712/713 exhibited the fastest labeling kinetics, with a reaction time clearly below 5 min, which likely reflects the high accessibility of this site for fluorophore attachment (Fig. 1g). The drop in $\tau$ of mCitrine associated with tet-Atto590 conjugation was reversed for all CONEGIs upon Atto590 photobleaching (Supplementary Fig. 1g, h). Furthermore, tet-Atto590 addition to cells co-expressing EGFR-QG-mCitrine and the BCNK incorporation system in presence of BCNK did not lead to an alteration in $\tau$ of mCitrine (Fig. 1g, h). This demonstrated that the obtained changes in $\tau$ in the CONEGIs result from specific FRET between mCitrine and Atto590 conjugated to site-specifically incorporated BCNK and not from tet-Atto590 non-specifically bound to EGFR or the PM, or from labeling of BCNK that might be incorporated at genomic amber codons. Washout of unbound tet-Atto590 after labeling of cells did not abolish the decrease in $\tau$ of mCitrine for the CONEGIs, confirming that the conjugation of tet-Atto590 to BCNK is stable (Supplementary Fig. 1i). The negative correlation of $\tau$ with the Atto590 fluorescence intensity and the saturation of the binding curves at high Atto590 concentrations for all CONEGIs further confirmed specificity of tet-Atto590-conjugation to BCNK in EGFR (Fig. 1h).

**CONEGI-851 reports on activation loop conformations**. To investigate whether the CONEGIs report on conformational changes in the TKD that occur upon activation, we measured $\tau$ of mCitrine at the PM by FLIM following stimulation with EGF to cells. We observed a significant decrease in $\tau$ of CONEGI-737, -843, and -851 and a significant increase in $\tau$ of CONEGI-712/713 upon EGF stimulation, whereas $\tau$ of CONEGI-713 and -730 did not significantly change (Fig. 2a; Supplementary Fig. 2a). To account for the variability in the completeness of the Atto590 labeling reaction that affects the initial FRET efficiency in a particular CONEGI construct, the difference in $\tau$ ($\Delta\tau$) relative to the $\tau$ before stimulation in each experiment was plotted (Fig. 2b). This $\Delta\tau$-time plot that follows the general trends of the $\tau$-time plot (Supplementary Fig. 2a) clearly reflects the conformational transitions of the CONEGI constructs. Neither $\tau$ of EGFR-QG-

mCitrine in presence of tet-Atto590 nor $\tau$ of EGFR(BCNK851)-QG-mCitrine in absence of tet-Atto590 did change upon EGF addition, precluding photophysical effects that change $\tau$ of mCitrine upon EGF stimulation (Fig. 2b; Supplementary Fig. 2b).

To examine whether the EGFR(BCNKXXX)-QG-mCitrine variants retain their activity upon BCNK incorporation, we quantified the phosphorylation on tyrosines 1086 ($Y_{1086}$) and 1148 ($Y_{1148}$) by measuring the interaction of mCherry-tagged phosphotyrosine-binding domain (PTB-mCherry) with each EGFR(BCNK)-QG-mCitrine variant upon EGF stimulation by FLIM[26]. EGF-induced PTB-mCherry recruitment to EGFR(BCNK843)-QG-mCitrine and EGFR(BCNK851)-QG-mCitrine at the PM resulted in a corresponding drop in $\tau$ (Fig. 2c, d). This decrease in $\tau$ was comparable to that of EGFR-QG-mCitrine upon EGF-mediated PTB-mCherry recruitment, whereas EGF-induced phosphorylation of other variants (712/713, 713, 730, and 737) only marginally increased as judged by minimal decreases in $\tau$ of mCitrine (Fig. 2d). To examine whether other phosphorylation sites were affected by BCNK incorporation and Atto590 labeling, we quantified the relative phosphorylation (pY/EGFR) on the Grb2-binding site $Y_{1068}$[27] for all CONEGIs by western blot analysis. Consistent with PTB-mCherry recruitment, CONEGI-843 and -851 exhibited a similar EGF-induced fold-change in $Y_{1068}$ phosphorylation as compared to EGFR-QG-mCitrine, but the relative phosphorylation level of CONEGI-843 was drastically reduced (Fig. 2e, f; Supplementary Fig. 2c). The other CONEGIs showed increased autonomous phosphorylation (CONEGI-712/713 and -713) or responded only marginally to EGF (CONEGI-712/713, -713, -730 and -737), indicating that their activation mechanism was impaired (Fig. 2e, f; Supplementary Fig. 2c). Immunofluorescence staining against pY$_{1068}$ in fixed HEK293T cells expressing EGFR(BCNK851)-QG-mCitrine showed that EGF-mediated phosphorylation at the PM was comparable to that of EGFR-QG-mCitrine (Fig. 2g; Supplementary Fig. 2d), which shows that EGFR(BCNK851)-QG-mCitrine at the PM is fully functional. Tet-Atto590 labeling did not affect autonomous or ligand-dependent EGFR phosphorylation of EGFR(BCNK851)-QG-mCitrine (Supplementary Fig. 2e). We therefore conclude that CONEGI-851 (from now on denoted as CONEGI) most closely follows the native activation mechanism of EGFR.

To investigate whether the change in FRET upon EGFR activation detected in CONEGI is of intra- or intermolecular origin, we measured $\tau$ of EGFR-QG-mCitrine (donor only) upon co-expression with an EGFR(BCNK851) variant that was labeled with Atto590 (acceptor only). $\tau$ of EGFR-QG-mCitrine in the presence of Atto590-labeled EGFR(BCNK851) ($2.992 \pm 0.006$ (SEM)) was close to that of EGFR-QG-mCitrine ($3.033 \pm 0.006$

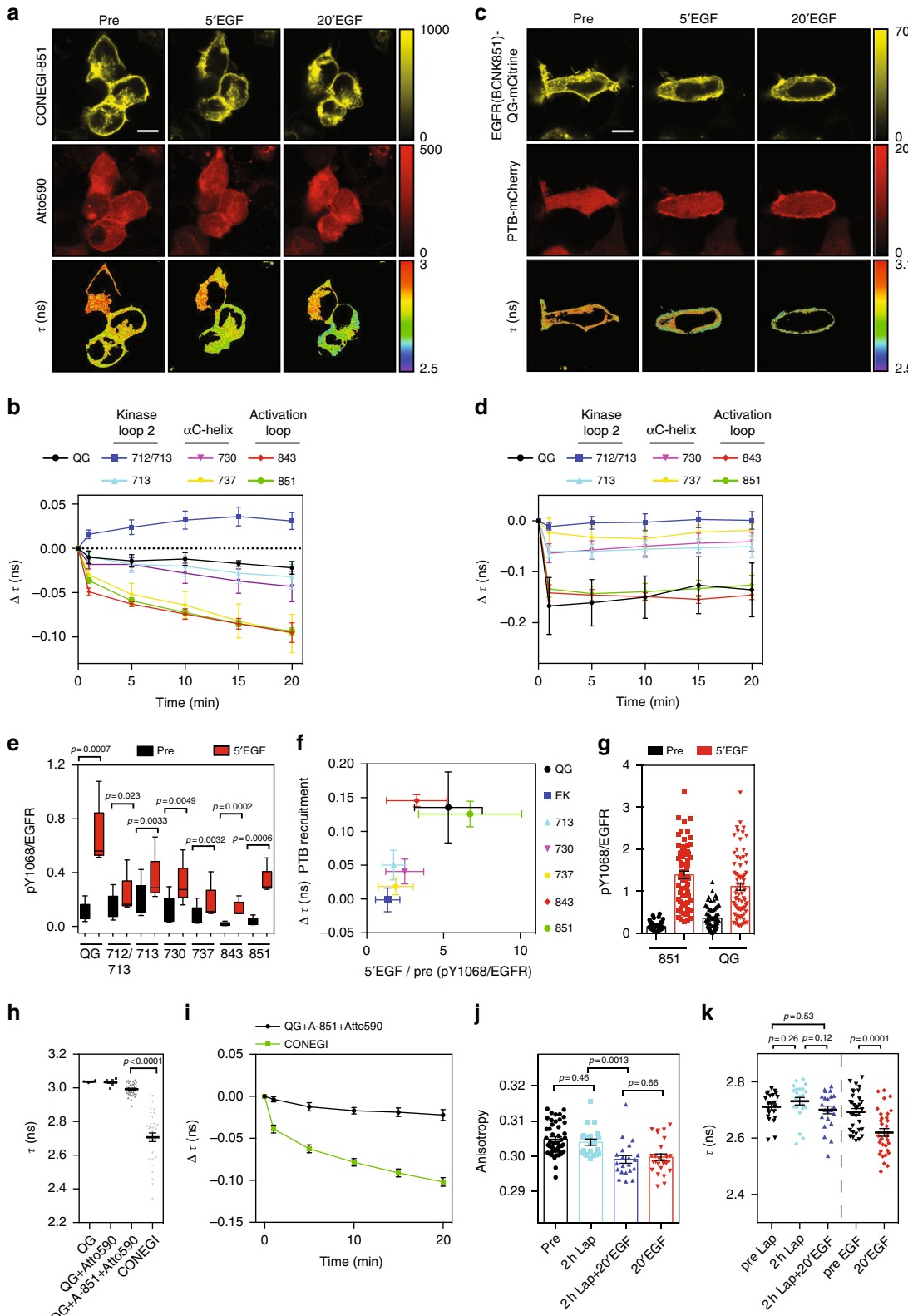

(SEM)) and only marginally decreased upon EGF stimulation (Fig. 2h, i; Supplementary Fig. 2f), showing that FRET in CONEGI reports on intramolecular conformational rearrangements. To demonstrate that the change in FRET efficiency in CONEGI originates from the rearrangement of the activation loop rather than from mCitrine reorientation upon dimerization, we locked

the TKD into an inactive conformation using the ATP analog EGFR inhibitor Lapatinib[28]. This abolished EGF-induced CON-EGI phosphorylation (Supplementary Fig. 2g), but did not prevent EGF-induced dimerization as measured by homo-FRET between mCitrine using fluorescence anisotropy[29] (Fig. 2j; Supplementary Fig. 2h). Importantly, this EGF-induced

**Fig. 2** CONEGI-851 reports on conformational changes in the activation loop. **a** Representative CONEGI-851 fluorescence and corresponding $\tau$ images upon EGF stimulation. **b** Change in $\Delta\tau$ of CONEGIs and EGFR-QG-mCitrine (QG: $n = 5$ cells; 712/713: $n = 7$; 713: $n = 7$; 730: $n = 7$; 737: $n = 7$; 843: $n = 6$; 851: $n = 31$) at the PM upon EGF stimulation (Supplementary Fig. 2a: absolute $\tau$). **c** Representative fluorescence images of EGFR(BCNK851)-QG-mCitrine, PTB-mCherry, and corresponding $\tau$ upon EGF stimulation. **d** Change in $\Delta\tau$ of EGFR-QG-mCitrine or EGFR(BCNKXXX)-QG-mCitrine variants (QG: $n = 4$ cells; 712/713: $n = 6$; 713: $n = 8$; 730: $n = 8$; 737: $n = 5$; 843: $n = 6$; 851: $n = 6$) at the PM upon EGF-mediated PTB-mCherry recruitment. **e** Relative $Y_{1068}$ phosphorylation ($pY_{1068}$/EGFR) of EGFR-QG-mCitrine and CONEGIs upon EGF stimulation by western blot analysis ($n = 5$; paired two-tailed $t$-test) (Supplementary Fig. 2c). **f** Fold-change in $Y_{1068}$ phosphorylation (5'EGF/pre) (quantified from **e**) versus $\Delta\tau$ upon PTB-mCherry recruitment (quantified from **d**) for CONEGIs and EGFR-QG-mCitrine. **g** Relative $Y_{1068}$ phosphorylation ($pY_{1068}$/EGFR) of EGFR-QG-mCitrine (QG) and EGFR(BCNK851)-QG-mCitrine (851) at the PM upon EGF stimulation by immunofluorescence (851: pre: $n = 72$ cells; 5'ÉGF: $n = 67$; QG: pre: $n = 73$; 5'ÉGF: $n = 74$) (Supplementary Fig. 2d). **h** $\tau$ of EGFR-QG-mCitrine in absence (QG; $n = 6$ cells) or presence of tetrazine-Atto590 (QG + Atto590; $n = 9$), EGFR-QG-mCitrine co-expressed with Atto590-labeled EGFR(BCNK851) (QG + A-851 + Atto590; $n = 36$) and CONEGI ($n = 32$; unpaired two-tailed $t$-test) (Supplementary Fig. 2f). **i** Change in $\Delta\tau$ of CONEGI ($n = 27$ cells) or EGFR-QG-mCitrine co-expressed with Atto590-labeled EGFR(BCNK851) ($n = 7$) upon EGF stimulation. **j** Mean mCitrine fluorescence anisotropy of CONEGI upon EGF stimulation in presence or absence of 1 µM Lapatinib (Lap) (Supplementary Fig. 2h) ($N = 3$ experiments; $n = 21–26$ fields of view/condition; unpaired two-tailed $t$-test). **k** $\tau$ of CONEGI at the PM upon EGF stimulation in presence (left; $n = 22$ cells) or absence (right; $n = 33$) of 1 µM Lap (unpaired two-tailed $t$-test). Scale bars, 10 µm. EGF stimulation, 100 ng ml$^{-1}$. Error bars: SEM, except Fig. 2e: Tukey box plot. $\tau$ fluorescence lifetime of mCitrine

dimerization of CONEGI with a Lapatinib locked TKD conformation did not result in a decrease in $\tau$ as observed for CONEGI in absence of inhibitor (Fig. 2k). This shows that CONEGI exclusively reports conformational transitions of the activation loop, but not reorientation of mCitrine upon dimerization.

**A catalytic conformation by activation loop phosphorylation.** Phosphorylation of $Y_{845}$ was previously described to affect both the conformation of the EGFR TKD, and to enhance its dimerization[9]. We therefore explored if $Y_{845}$ phosphorylation in the activation loop is linked to an EGFR conformational state that is catalytically active without dimerization. For this, we compared the dimerization and conformational transition of CONEGI upon its ligand-independent phosphorylation to that of a CONEGI-Y845F mutant. Ligand-independent phosphorylation of EGFR is known to occur upon phosphatase inhibition by pervanadate (PV)[30] or upon high EGFR expression levels[14,15,31,32]. Under both conditions, the intrinsic EGFR kinase activity surpasses the reverse dephosphorylating activity of phosphatases, resulting in an increased steady state phosphorylation level of EGFR[15].

EGF administration at saturating dose (100 ng ml$^{-1}$) resulted in a drop in anisotropy of CONEGI (independent of UAA incorporation or Atto590 labeling), showing EGF-induced dimerization (Fig. 3a; Supplementary Fig. 3a, b). This dimerization-induced drop in anisotropy could be affected by the efficiency of UAA incorporation in response to the amber stop codon, resulting in the expression of truncated CONEGI (~43 ± 0.4% (SEM), Supplementary Fig. 3c). This truncated CONEGI lacks the donor fluorophore mCitrine, but potentially could interact with full length CONEGI. We therefore measured the amount of full length CONEGI at the PM by determining the fluorescence ratio of EGF-Alexa647 (100 ng ml$^{-1}$, binds to all CONEGI constructs) to mCitrine (only expressed on full length CONEGI) (Supplementary Fig. 3d). The identical fluorescence ratio of EGFR(BCNK851)-QG-mCitrine and full length EGFR-QG-mCitrine (Supplementary Fig. 3d) showed that truncated CONEGI did not reach the PM. The identical anisotropy responses to EGF of EGFR-QG-mCitrine and unlabeled EGFR (BCNK851)-QG-mCitrine also confirmed that only full length CONEGI was present at the PM (Supplementary Fig. 3a). The relatively smaller EGF-induced change in anisotropy for CONEGI can be accounted for by the ~14.3 ± 0.8% (SEM) reduction in mCitrine fluorescence lifetime by FRET to Atto590.

EGF stimulation led to a concurrent increase in phosphorylation of $Y_{1068}$ and $Y_{845}$ (Fig. 3b, c; Supplementary Fig. 3e, f). In contrast, the anisotropy of CONEGI did not change upon PV treatment over a time course of 20 min, in which $Y_{1068}$ and $Y_{845}$ phosphorylation reached similar levels as compared to EGF stimulation (Fig. 3a–c; Supplementary Fig. 3a, e, f). This demonstrates that the majority of ligand-independent phosphorylated CONEGI is monomeric[15,33]. Both EGF and PV induced a conformational change in CONEGI as apparent from the decrease in $\tau$. However, while EGF provoked a rapid drop in $\tau$, PV treatment resulted in a slower decrease in agreement with the differential phosphorylation kinetics (Fig. 3b–d; Supplementary Fig. 3e–g). This shows that not only EGF-stimulated dimers but also phosphorylated monomers undergo a conformational change in the activation loop upon phosphorylation. For CONEGI-Y845F expressing HEK293T cells, a significantly smaller and slower change in $\tau$ upon PV was observed showing that Y845 phosphorylation induces a conformational change in the activation loop (Fig. 3f; Supplementary Fig. 3h). This was paralleled by a significantly lower $Y_{1068}$ auto-phosphorylation in CONEGI-Y845F as compared to that of CONEGI, revealing that $Y_{845}$ phosphorylation results in a catalytically more active CONEGI conformation (Fig. 3e; Supplementary Fig. 3i).

We next investigated the dependence of ligand-independent phosphorylation on the expression level of CONEGI in relation to the conformational transition of the activation loop. Exploiting the cell-to-cell variance in mCitrine fluorescence, allowed us to relate CONEGI expression to phosphorylation on $Y_{1068}$ as measured by immunofluorescence. We could clearly observe an expression level-dependent increase in auto-phosphorylation that was absent in its Y845F mutant (Fig. 3g, i; Supplementary Fig. 3j). This increase in auto-phosphorylation was not due to an increase in dimers at higher expression as apparent from the fluorescence anisotropy as function of CONEGI expression (Supplementary Fig. 3a). In an analogous experiment, we related CONEGI expression by its fluorescence intensity to its conformational state by fluorescence lifetime in many individual cells. Here, we could clearly observe a CONEGI expression level-dependent conformational transition that was absent in the CONEGI-Y845F mutant (Fig. 3h, i). Together, these data show that CONEGI but not its Y845F mutant is able to activate other CONEGI molecules via $Y_{845}$ phosphorylation, resulting in catalytic amplification. In order to estimate at which EGFR expression levels this spontaneous amplification in EGFR phosphorylation can occur, we determined the level of CONEGI expression in relation to endogenous EGFR expression in non-

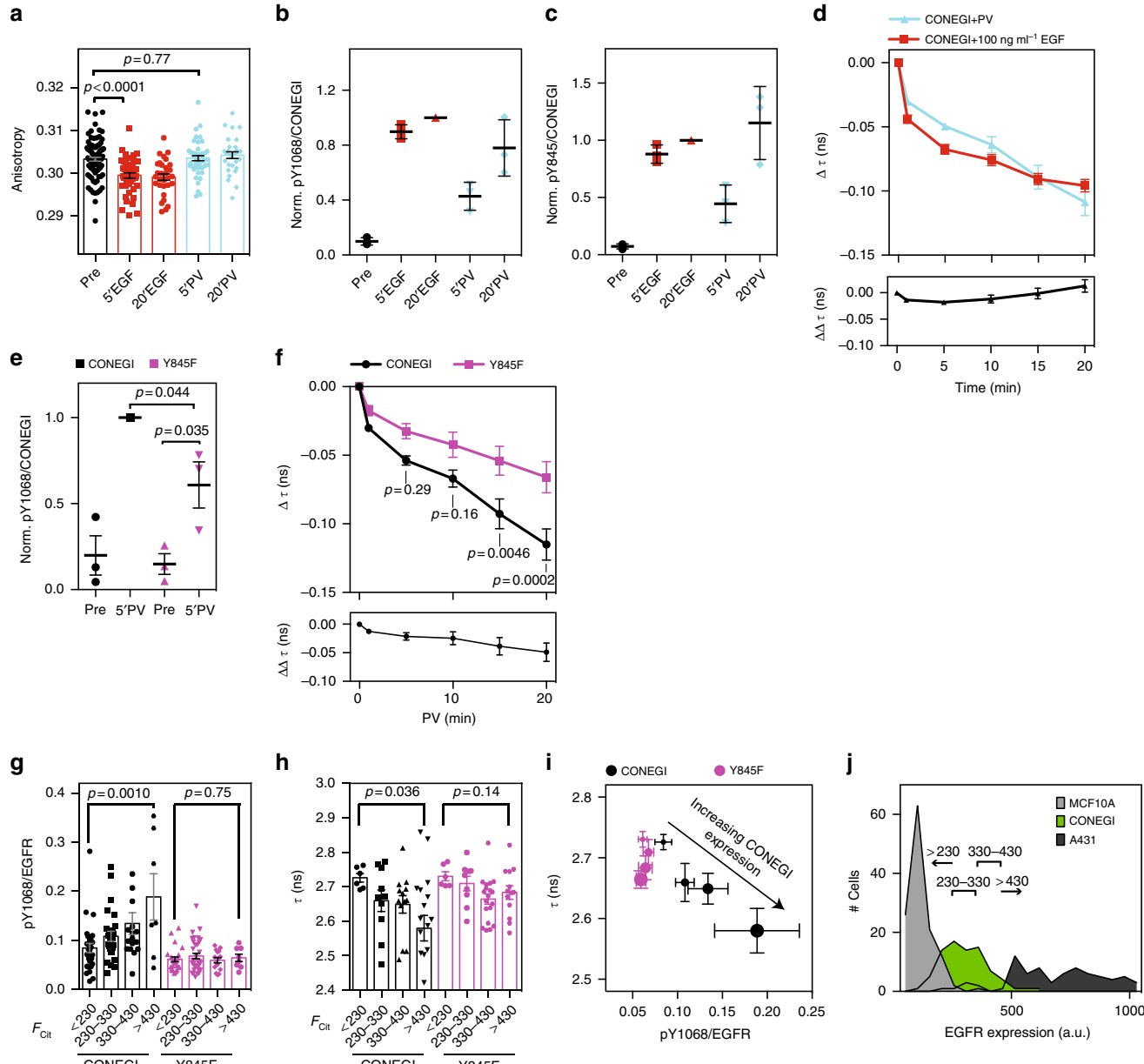

**Fig. 3** $Y_{845}$ phosphorylation induces an active conformation of EGFR monomers. **a** Mean mCitrine fluorescence anisotropy of CONEGI upon EGF or pervanadate (PV) treatment (Supplementary Fig. 3a, b) ($N = 3$ experiments; $n = 30$–47 fields of view/condition; unpaired two-tailed $t$-test). **b, c** Normalized relative $Y_{1068}$ (pY1068/CONEGI) (**b**) or pY845 phosphorylation (pY845/CONEGI) (**c**) of CONEGI upon EGF or PV treatment by western blot analysis ($n = 3$) (Supplementary Fig. 3e, f). **d** Change in $\Delta\tau$ of CONEGI (upper panel) at the PM and the difference between EGF and PV treatment ($\Delta\Delta\tau$) (lower panel) (EGF: $n = 19$ cells; PV: $n = 6$) (Supplementary Fig. 3g). **e** Normalized relative $Y_{1068}$ phosphorylation (pY1068/CONEGI) of CONEGI or CONEGI-Y845F upon PV treatment by western blot analysis ($n = 3$; unpaired two-tailed $t$-test) (Supplementary Fig. 3i). **f** Change in $\Delta\tau$ of CONEGI and CONEGI-Y845F upon PV treatment (upper panel) and the difference ($\Delta\Delta\tau$) between CONEGI and CONEGI-Y845F (lower panel) (CONEGI: $n = 6$ cells; Y845F: $n = 8$; two-way ANOVA) (Supplementary Fig. 3h). **g** Relative $Y_{1068}$ phosphorylation (pY1068/EGFR) of CONEGI and CONEGI-Y845F at increasing EGFR expression levels as measured by mCitrine intensity per cell ($F_{cit}$ binned as follows: <230, 230–330, 330–430, >430) (CONEGI: $n = 80$ cells; Y845F: $n = 81$; unpaired two-tailed $t$-test). **h** Mean $\tau$ in CONEGI and CONEGI-Y845F at increasing EGFR expression levels ($n = 47$ cells/variant; unpaired two-tailed $t$-test). **i** Relative $Y_{1068}$ phosphorylation (pY1068/EGFR) of CONEGI and CONEGI-Y845F versus their $\tau$ of mCitrine upon increasing expression levels ($F_{cit} < 230$, 230–330, 330–430, >430; as indicated by increasing dot size). **j** Distribution of relative EGFR expression in MCF10A ($n = 125$ cells), A431 ($n = 129$), and CONEGI-expressing HEK293T cells ($n = 80$) as derived from fluorescence histograms of EGF-Alexa647 binding per cell to the respective cell lines (Supplementary Figure 3k). EGF stimulation, 100 ng ml$^{-1}$. PV treatment, 0,33 mM. Error bars: SEM. $\tau$ fluorescence lifetime of mCitrine

tumorigenic epithelial MCF10A (~$1 \times 10^5$ receptors/cell) and epidermoid carcinoma A431 cells (~$1 \times 10^6$ receptors/cell)[34–36]. Comparison of EGF-Alexa647 binding to CONEGI in HEK293T cells to that of endogenous EGFR in A431 and

MCF10A cells revealed that between ~$2 \times 10^5$–$5 \times 10^5$ CONEGIs are expressed per cell (Supplementary Fig. 3k). Matching the intensity distribution of the EGF-Alexa647 fluorescence bound to CONEGI to the distribution of mCitrine intensities of

CONEGI obtained from the pY1068 immunofluorescence experiments, allowed us to estimate the CONEGI expression level from mCitrine fluorescence. This showed that spontaneous phosphorylation occurs at all CONEGI expression levels but strongly increases at $3.8 \times 10^5$–$4.6 \times 10^5$ receptors per cell, which falls within the lower range of endogenous EGFR expression in A431 cells (Fig. 3j).

**EGFR dimers can induce autocatalytic activation of monomers**. We next investigated whether ligand-induced EGFR dimers can activate receptor monomers. At saturating EGF dose (100 ng ml$^{-1}$), all receptors are occupied with ligand and are activated by the canonical dimerization mechanism. However, catalytic amplification can take place at sub-saturating EGF dose (20 ng ml$^{-1}$) because only a fraction of receptors will be occupied by ligand (20–30%; estimated by the ratio of EGF-Alexa647/mCitrine fluorescence at 20 ng/ml over 100 ng/ml EGF-Alexa647 per cell) (Supplementary Fig. 4a). We therefore compared CONEGI and CONEGI-Y845F phosphorylation and conformational dynamics in cells that were stimulated with saturating or sub-saturating EGF dose. Upon stimulation with saturating EGF dose, CONEGI and CONEGI-Y845F exhibited similar rapid $Y_{1068}$ phosphorylation reaching comparable levels (Fig. 4a; Supplementary Fig. 4b). CONEGI and CONEGI-Y845F also exhibited similar activation loop dynamics as apparent from the comparable change in $\tau$ over time (Fig. 4b; Supplementary Fig. 4d). This is consistent with the activation loop being rearranged to an open conformation in both CONEGI and its Y845F mutant by the canonical allosteric dimerization mechanism. A similar EGF-induced decrease in anisotropy for CONEGI and CONEGI-Y845F confirmed that Y845F mutation does not affect its dimerization (Supplementary Fig. 4c).

However, upon stimulation of cells with sub-saturating EGF dose, $Y_{1068}$ phosphorylation of CONEGI reached within 1 min $\sim39 \pm 9\%$ (SEM) of that at saturating EGF dose to further increase over time to $\sim72 \pm 12\%$ (SEM), showing a clear amplification of $Y_{1068}$ phosphorylation over time. This amplification was absent in CONEGI-Y845F, which after rapid initial activation remained stable at $\sim49 \pm 7\%$ (SEM) of that of CONEGI at saturating EGF (Fig. 4c; Supplementary Fig. 4b). Strikingly, the change in $\tau$ of CONEGI-Y845F upon sub-saturating EGF stimulation was significantly slower and of lesser magnitude as compared to that of CONEGI, which was comparable to that at saturating EGF (Fig. 4d; Supplementary Fig. 4e). This shows that EGF-activated CONEGI dimers induce an active conformation in unliganded CONEGI monomers by $Y_{845}$ phosphorylation (Fig. 4e; Supplementary Fig. 4f). The question remained if $Y_{845}$ is directly phosphorylated by the intrinsic kinase activity of EGFR or whether this happens indirectly by recruitment or activation of another tyrosine kinase that is dependent on C-terminal tyrosine auto-phosphorylation of EGFR. It was previously shown that Y to A mutation of the $Y_{1086}$ in the C-terminal tail strongly suppressed $Y_{845}$ phosphorylation[37]. To address if $Y_{845}$ phosphorylation happens via direct trans-phosphorylation or indirectly via recruitment of a tyrosine kinase to the phosphorylated C-terminal tail, we investigated $Y_{845}$ phosphorylation and conformational dynamics of a C-terminal tail truncated CONEGI mutant (CONEGI-ΔD969). This mutant retained its ability to dimerize (Supplementary Fig. 4g), but was severely impaired in $Y_{845}$ phosphorylation (Fig. 4f, Supplementary Fig. 4h). At saturating EGF dose, CONEGI-ΔD969 followed closely the conformational dynamics of CONEGI (Fig. 4g; Supplementary Fig. 4i), consistent with activation by the allosteric dimerization mechanism. However, at sub-saturating EGF dose, the change in $\tau$ of CONEGI-ΔD969 was significantly slower and of lesser

magnitude as compared to that of CONEGI (Fig. 4h; Supplementary Fig. 4i). In fact, C-terminal truncation led to a similar attenuation of the conformational transition as the Y845F mutation (compare Fig. 4h to 4d). This shows that C-terminal auto-phosphorylation on EGFR is necessary to recruit or activate another tyrosine kinase that activates monomers by trans-phosphorylation on $Y_{845}$[37]. These activated monomers can phosphorylate EGFR on the C-terminal tail, closing an auto-catalytic loop.

**Discussion**
Many insights in the ligand-induced allosteric dimer activation mechanism of EGFR were gained by structural studies that captured different conformations of the ECD and the TKD[2,5]. On the other hand, genetically encoded optical kinase substrate sensors provided information about spatially resolved EGFR phosphorylation dynamics[26,38], but could not relate EGFR conformational dynamics to its phosphorylation activity. To bridge this gap, we developed the conformational sensor CONEGI that reports on conformational transitions in the activation loop of the EGFR TKD in living cells, thereby enabling us to relate EGFR's conformation to its phosphorylation activity and dimerization.

We designed multiple CONEGI variants in a hybrid approach using genetic code expansion in order to detect conformational dynamics in different key functional regions of the TKD. The limited availability of membrane-permeable dyes restricted the selection of an acceptor fluorophore with the spectral properties to form a FRET pair with mCitrine. This likely explains why existing conformational, FRET-based EGFR sensors are either based on protein labeling of the ECD[39,40] or on in vitro labeling of intracellular domains[41]. The development of approaches to deliver membrane-impermeable dyes into cells[42] or of novel membrane-permeable fluorophore species, e.g., near-infrared silicon-rhodamine dyes[43] will facilitate the generation of similar conformational FRET sensors in the future. Here, we proved BCNK-labeling with the membrane-permeable Atto590 to be stable and highly specific in living cells (Fig. 1d, e, g, h). BCNK incorporation at the individual sites of the EGFR TKD affected kinase functionality to a different extent. Increased autonomous phosphorylation of CONEGI-712/713 and -713 might be caused by perturbation of inhibitory electrostatic interactions of kinase loop 2 with the PM[7]. The low EGF-induced fold-change in phosphorylation of CONEGI-737 could originate from hindrance of salt bridge formation between $K_{721}$ and $E_{738}$ that is essential for adopting the active conformation[9] or from perturbation of the αC-helix structure. In contrast, BCNK incorporation at position 851 in the activation loop was compatible with kinase function probably because the activation loop is a highly flexible region and the incorporation site is sufficiently apart from the catalytically important DFG motif (830–832), $Y_{845}$ and a β-strand that turns into a two-turn helix upon activation[10].

With CONEGI, we investigated if and by which mechanism catalytic EGFR activation occurs. The comparison in dimerization (by fluorescence anisotropy) and conformational transitions in the activation loop (by FLIM) of CONEGI with its Y845F mutant, not only showed the conformational transition driven by allosteric interactions in EGF-activated EGFR dimers[2], but also that $Y_{845}$ phosphorylation induces a conformational change in EGFR monomers. That this conformational transition induced by $Y_{845}$ phosphorylation leads to a catalytically more active conformation was apparent from the increased $Y_{1068}$ auto-phosphorylation on CONEGI monomers with respect to the CONEGI–Y845F mutant (Fig. 3e–i).

Oligomerization at sub-saturating EGF dose has been proposed to amplify receptor phosphorylation beyond what is

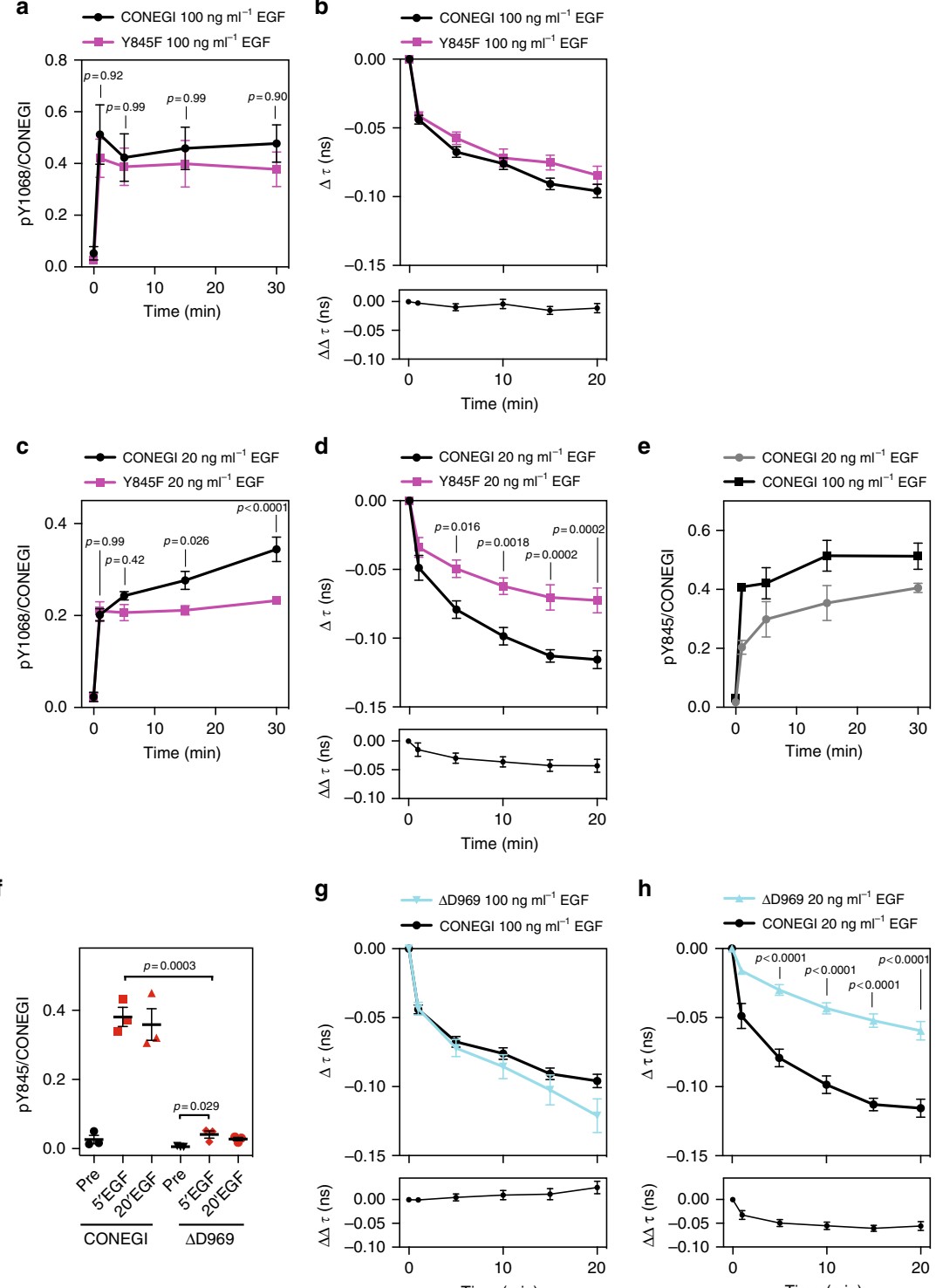

expected from liganded EGFR dimers only[44]. In principle, this could explain the amplification of phosphorylation that we ascribe to autocatalysis. However, under these conditions, we detected an equivalent decrease in anisotropy for CONEGI and its Y845F mutant showing that they exhibited identical self-association levels. While the dimerization level of the CONEGI Y845F mutant correlated with half-maximum $Y_{1068}$ phosphorylation with respect to saturating EGF dose, $Y_{1068}$ phosphorylation of CONEGI reached ~72% of that at saturating EGF dose (Fig. 4a, c). This difference of ~22% in $Y_{1068}$ phosphorylation showed that CONEGI dimers can

phosphorylate CONEGI monomers. Furthermore, we could also show that at high CONEGI expression levels or inhibition of phosphatases by PV, $Y_{845}$ phosphorylation induced a catalytically active conformation by displacing the activation loop that enabled trans-phosphorylation between EGFR monomers on $Y_{1068}$. The PV-induced residual conformational transition in CONEGI-Y845F that correlated with a lower $Y_{1068}$ phosphorylation suggests that besides $Y_{845}$ phosphorylation, additional posttranslational reactions such as ubiquitination or interactions with other proteins could contribute to generate a catalytically competent conformation

**Fig. 4** EGFR dimers can induce autocatalytic activation of EGFR monomers. **a** Relative $Y_{1068}$ phosphorylation ($pY_{1068}$/CONEGI) of CONEGI and CONEGI-Y845F upon stimulation with 100 ng ml$^{-1}$ EGF by western blot analysis ($n = 4$; two-way ANOVA) (Supplementary Fig. 4b). **b** Change in $\Delta\tau$ of CONEGI and CONEGI-Y845F at the PM (upper panel) upon stimulation with 100 ng ml$^{-1}$ EGF and the difference ($\Delta\Delta\tau$) between CONEGI and CONEGI-Y845F (lower panel) (CONEGI: $n = 19$ cells; Y845F: $n = 8$) (Supplementary Fig. 4d). **c** Relative $Y_{1068}$ phosphorylation ($pY_{1068}$/CONEGI) of CONEGI and CONEGI-Y845F upon stimulation with 20 ng ml$^{-1}$ EGF by western blot analysis ($n = 4$; two-way ANOVA) (Supplementary Fig. 4b). **d** Change in $\Delta\tau$ of CONEGI and CONEGI-Y845F at the PM upon stimulation with 20 ng ml$^{-1}$ EGF (upper panel) and the difference ($\Delta\Delta\tau$) between CONEGI and CONEGI-Y845F (lower panel) (CONEGI: $n = 6$ cells; Y845F: $n = 7$; two-way ANOVA) (Supplementary Fig. 4e). **e** Relative $Y_{845}$ phosphorylation ($pY_{845}$/CONEGI) of CONEGI upon stimulation with 20 and 100 ng ml$^{-1}$ EGF by western blot analysis ($n = 3$) (Supplementary Fig. 4f). **f** Relative $Y_{845}$ phosphorylation ($pY_{845}$/CONEGI) of CONEGI and CONEGI-$\Delta$D969 upon stimulation with 100 ng ml$^{-1}$ EGF by western blot analysis ($n = 3$; unpaired two-tailed $t$-test) (Supplementary Fig. 4h). **g**, **h** Change in $\Delta\tau$ of CONEGI-$\Delta$D969 and CONEGI at the PM upon stimulation with 20 (**h**) and 100 ng ml$^{-1}$ EGF (**g**) (upper panel) and the difference ($\Delta\Delta\tau$) between CONEGI and CONEGI-$\Delta$D969 (lower panel) (CONEGI: 20 ng ml$^{-1}$ EGF: $n = 6$ cells; 100 ng ml$^{-1}$ EGF: $n = 19$; $\Delta$D969: 20 ng ml$^{-1}$ EGF: $n = 9$; 100 ng ml$^{-1}$: $n = 6$; two-way ANOVA) (Supplementary Fig. 4i). Error bars: SEM. $\tau$ fluorescence lifetime of mCitrine

in EGFR monomers. BCNK incorporation at the known ubiquitination site $K_{851}$[45] did not alter the phosphorylation activity of CONEGI (Fig. 2d, e, g), but other ubiquitination sites (e.g., $K_{843}$) close to the EGFR activation loop could affect the conformation of the activation loop and thereby its kinase activity. However, we previously showed that EGFR monomers are inefficiently ubiquitinated[15] in line with $Y_{845}$ phosphorylation, being the dominant mechanism for EGFR phosphorylation amplification.

The diminished $Y_{845}$ phosphorylation and impaired conformational transition in C-terminal tail truncated CONEGI-$\Delta$D969 monomers, further showed that these depend on autophosphorylation of the C-terminal tail (Fig. 4f, h). EGFR autocatalysis can therefore occur by an $Y_{845}$ phosphorylation induced catalytically active conformation in EGFR monomers that transphosphorylate EGFR monomers on their C-terminal tail. These auto-phosphorylation events on the C-terminal tail then induce the recruitment or activation of a tyrosine kinase that phosphorylates $Y_{845}$ on EGFR monomers, thereby generating a looped causality that amplifies EGFR phosphorylation. Phosphorylation of the C-terminal tail by active Y845 phosphorylated monomeric receptors as well as the phosphorylation of $Y_{845}$ by C-terminal tail-associated tyrosine kinases, indicates that the flexible C-terminal tail can reach the active site of another EGFR monomer, thereby overcoming the steric hindrance of the tethered inhibitory conformation of the ECD[5]. Src is known to be activated by C-terminally phosphorylated EGFR and capable of phosphorylating $Y_{845}$[46,47] and therefore likely involved in this autocatalytic amplification on EGFR.

Theoretically, the autocatalytic phosphorylation mechanism can generate only one collective state in which all receptors are fully active[48]. Therefore, the question arises how autocatalytic activation is regulated by interaction with other enzymatic activities. We previously demonstrated that continuous vesicular recycling of EGFR monomers through perinuclear areas with high protein tyrosine phosphatase (PTP) activity of ER-associated PTPs, counteracts spontaneous autocatalytic EGFR activation by $Y_{845}$ dephosphorylation on recycling receptors[15]. The spatial separation of PTP activity from EGFR activity at the PM thereby provides a means via which cells can suppress autonomous autocatalytic activation while maintaining responsiveness to EGF. The collective response properties of EGFR to EGF at the PM however arise from the autocatalytic property of EGFR in conjunction with recursive interaction with PTPRG in the form of a double negative feedback[14,36,48,49].

Autocatalytic EGFR activation can lead to full EGFR phosphorylation at the PM above a threshold EGF concentration[14,50,51]. However, EGFR is organized in clusters of 100–200 nm in diameter with up to 250 EGFR monomers[33,52,53] limiting the reach of autocatalysis. This could effectively create

nanoscopic systems that can locally sense and robustly respond to growth factors.

## Methods

**Antibodies.** Rabbit anti-EGFR (western blot 1:1000; 4267, Cell Signaling Technology, Danvers, MA), goat anti-EGFR (western blot 1:1000; AF231, R&D Systems, Minneapolis, MN), mouse anti-pY1068 (western blot: 1:1000; immunofluorescence: 1:200; 2236, Cell Signaling Technology), mouse anti-pY845 (western blot 1:1000; 558381; BD Biosciences, Heidelberg, Germany), mouse anti-phosphotyrosine (PY72) (western blot 1:730; P172.1, InVivo Biotech Services, Henningsdorf, Germany), mouse monoclonal anti-$\alpha$-Tubulin (western blot 1:4000; Sigma-Aldrich, St. Louis, MO), rabbit anti-GAPDH (western blot 1:1000; 2118, Cell Signaling Technology), living colors rabbit anti-GFP (western blot 1:1000; 632593, Clontech, Mountain View, CA), IRDye 680 donkey anti-mouse IgG (western blot 1:10,000; LI-COR Biosciences, Lincoln, NE), IRDye 800 donkey anti-rabbit IgG (western blot 1:10,000; LI-COR Biosciences), Alexa Fluor® 647 donkey anti-mouse IgG (immunofluorescence 1:200; Thermo Fisher Scientific Inc., Waltham, MA).

**Plasmids.** Restriction and ligation enzymes were purchased from New England Biolabs (NEB, Frankfurt am Main, Germany). mCitrine-N1, mCherry-N1, and mTurquoiseN1 were generated by insertion of AgeI/BsrgI PCR fragments of mCitrine, mCherry, and mTurquoise cDNA in pEGFP-C1 (Clontech, Mountain View, CA). Fusions of EGFR and PTB domain with fluorescent proteins were generated through restriction-ligation of EGFR and PTB domain cDNA into the appropriate vector. To generate EGFR-QG-mCitrine, mCitrine flanked with a linker sequence (LAAAYSSILSSNLSSDS-mCitrine-SDSSLNSSLISSYAAAL) was inserted between Q958 and G959 of EGFR. Site-directed mutagenesis PCR using *PfuUltra* high-fidelity DNA polymerase (Agilent Technologies, Santa Clara, CA) replaced coding sequences in EGFR-QG-mCitrine with an amber codon (TAG) to generate BCNK incorporation sites and was also used to generate the CONEGI-Y845F mutant. NotI/NheI EGFR(TAG)-QG-mCitrine PCR fragments were inserted in a (U6-PylT*)$_4$/EF1$\alpha$ plasmid previously described in ref. [23]. The BCNK incorporation system consists of two plasmids, (U6-PylT*)$_4$/EF1$\alpha$-PylRS to express the tRNA synthetase and peRF1(E55D) for expression of a modified eukaryotic release factor 1 (eRF1). peRF1(E55D) was described earlier[23] and (U6-PylT*)$_4$/EF1$\alpha$-PylRS was modified by adding a nuclear export sequence (NES) LALKLAGLDIGS attached via a flexible 4× (SGGGGS) linker to the N-terminus of PylRS. The plasmid was created from a previously reported construct[54] by inverse PCR followed by insertion of the 4× (SGGGGS) linker via blunt end ligation. A complete list with all relevant primers is provided in Supplementary Table 2. All constructs were sequence verified and tested for correct expression.

**Reagents.** Human EGF (Peprotech, Hamburg, Germany) was shock frozen at a concentration of 100 µg ml$^{-1}$ in PBS + 0.1% BSA and stored at −80 °C. Pervanadate was freshly prepared by adding sodium orthovanadate (S6508, Sigma-Aldrich) to H$_2$O$_2$ (30%) according to ref. [30]. BCNK was described earlier in ref. [24]. Site-specific C-terminal labeling of hEGF with Alexa647-maleimide (Life Technologies, Darmstadt, Germany) was carried out as described in ref. [55]. Briefly, the His-CBD-Intein-(Cys)-hEGF-(Cys) plasmid (kindly provided by Professor Luc Brunsveld, University of Technology, Eindhoven), was expressed in *Escherichia coli* BL21 (DE3) and protein was purified by Ni-NTA affinity chromatography from inclusion bodies. Purified protein was refolded in vitro, followed by intein splicing and size exclusion chromatography. This purified human EGF was then C-terminally labeled with a fivefold molar excess of Alexa647-maleimide, excess dye was removed by dialysis, and the labeled protein stored in PBS at −20 °C. Tet-Atto590 was synthesized by conjugating Atto590-NHS ester (79636, Sigma-Aldrich) to tet-NH$_2$ in the presence of N,N-Diisopropylethylamine (DIPEA). The product was purified using high-performance liquid chromatography (HPLC) and

the identity of the compound confirmed by mass spectroscopy. Lapatinib (231277-92-2, Cayman Chemical, Ann Arbor, MI) was solubilized in ethanol to a stock concentration of 10 mM and stored at 4 °C.

**Crystal structure**. PDB files of crystal structures of the active and inactive EGFR TKD were downloaded from the RSCB protein data bank. Alignment of the crystal structures and measurement of the distances between insertion sites were performed using the program MacPyMOL (http://www.pymol.org).

**Cell culture and site-specific labeling**. Identities of HEK293T (ATCC: CRL-11268), MCF10A (ATCC: CRL-10317), and A431 (DSMZ: ACC91) cells were determined by DNA profiling using eight different and highly polymorphic short tandem repeat (STR) loci and testing for the presence of mitochondrial DNA sequences from rodent cells as mouse, rat, chinese and syrian hamster (DSMZ) and tested regularly for mycoplasma contamination using MycoAlert Mycoplasma detection kit (Lonza). MCF10A cells were grown in DMEM/F12 media supplemented with 5% horse serum, 20 ng ml$^{-1}$ EGF, 0.5 µg ml$^{-1}$ hydrocortisone (H-0888, Sigma-Aldrich), 100 ng ml$^{-1}$ cholera toxin (Sigma-Aldrich), 10 µg ml$^{-1}$ insulin (Sigma-Aldrich), and 1% glutamine and maintained at 37 °C in 5% CO$_2$. A431 and HEK293T cells were grown in Dulbecco's Modified Eagle's Medium (DMEM) supplemented with 10% fetal bovine serum, 2 mM L-glutamine and 1% non-essential amino acids (NEAAs) and maintained at 37 °C in 5% CO$_2$. Transfection of mammalian cells with plasmid DNA was achieved using Fugene®6 according to the manufacturers protocol. HEK293T cells in one well of an eight-well Labtek (for confocal microscopy and FLIM) were transfected with 250 ng of the EGFR expression plasmid, 100 ng (U6-PylT*)$_4$/EF1α-PylRS and 100 ng peRF1 (E55D). Cells in one well of a 24-well dish (for western blot analysis) were transfected with 500 ng of the EGFR expression plasmid, 100 ng (U6-PylT*)$_4$/EF1α-PylRS, and 100 ng peRF1(E55D). HEK293T cells in MatTeks (for anisotropy) were transfected with 1000 ng of the EGFR expression plasmid, 100 ng (U6-PylT*)$_4$/EF1α-PylRS, and 100 ng peRF1(E55D). Media was replaced by fresh media supplemented with 1 mM BCNK before addition of the transfection mix to cells. Cells were incubated overnight at 37 °C with 5% CO$_2$ and BCNK was washed out on the next day. Labeling was performed using 400 nM tet-Atto590 in DMEM for 20 min at room temperature (RT). The washout of free diffusing tet-Atto590 was carried out for at least 2 h with media exchange every 15–20 min.

**Western blotting and in-gel fluorescence**. Cells were lysed in ready-made cell lysis buffer (9803, Cell Signaling Technology) supplemented with Complete Mini EDTA-free protease inhibitor (Roche Applied Science, Heidelberg, Germany) and 100 µl phosphatase inhibitor cocktail 2 and 3 (P5726 and P0044, Sigma-Aldrich). Following lysis, samples were cleared by centrifugation for 10 min, 13,000 r.p.m. at 4 °C. Bis-Tris-PAGE was performed using the X-cell II mini electrophoresis apparatus (Life Technologies) according to the manufacturer's instructions. Proteins were transferred to preactivated polyvinylidene difluoride (PVDF) membranes (Merck Chemicals, Darmstadt, Germany) and incubated with the respective primary antibodies at 4 °C overnight. Detection was performed using species-specific IR-Dye 800 CW and IR-Dye 680 secondary antibodies (LI-COR Biosciences, Bad Homburg vor der Höhe, Germany) and the Odyssey Infrared Imaging System (LI-COR Biosciences). Eventually, Bis-Tris gels were imaged with the Typhoon Trio Variable Mode Imager (GE Healthcare, Buckinghamshire, UK) to detect fluorescently labeled proteins before western blotting. Atto590 was excited using a 532 nm laser and fluorescence emission was detected with a 610/30 BP filter at a resolution of 100 µm. Uncropped western blots and in-gel fluorescence images are shown in Supplementary Fig. 5.

**Immunofluorescence**. Cells were fixed with 4% paraformaldehyde (Roti®-Histofix 4%, Carl Roth GmbH, Karlsruhe, Germany) for 10 min at RT and permeabilized for 5 min with 0.1% Triton X-100 in TBS. Background staining was blocked by incubation with Odyssey® Blocking Buffer (LI-COR Biosciences) for 1 h at 4 °C. Primary antibodies diluted in Odyssey® Blocking Buffer were applied overnight at 4 °C and secondary antibodies for 1 h at RT. Fixed cells were imaged in TBS at 37 °C. In background-subtracted images, masks for the PM of single cells were generated and the mean fluorescence intensity for each channel was measured in ImageJ (http://imagej.nih.gov/ij/). The relative phosphorylation on Y$_{1068}$ was determined per cell and the mean values of Y$_{1068}$/EGFR-QG-mCitrine variants were calculated.

**Fluorescence microscopy**. Confocal images at the Olympus FV1000 equipped with a 60×/1.35 numerical aperture Oil UPLSApo objective (Olympus, Hamburg, Germany) and a temperature controlled incubation chamber (EMBL, Heidelberg, Germany) were acquired in sequential mode frame by frame with 2× line averaging. The pinhole was set to 2.5 airy units. mCitrine was excited with a 488 nm Argon laser (GLG 3135, Showa Optronics, Tokyo, Japan), mCherry/Atto590 with a 561 nm DPPS laser (85-YCA-020-230, Melles Griot, Bensheim, Germany), and Alexa647 with a 633 HeNe laser (05LHP-991, Melles Griot, Bensheim, Germany) using a DM405/488/561/633 dichroic mirror. mCitrine fluorescence was detected between 498 and 551 nm using the SDM560 beam splitter. Atto590/mCherry fluorescence was detected in the bandwidth of 575–675 nm and Alexa647 fluorescence between 643 and 743 nm. With these settings, no bleed through between

mCitrine and Atto590 or mCherry was observed. Live cells were imaged in imaging media at 37 °C and 5% CO$_2$ and stimulated with 20 ng ml$^{-1}$ EGF, 100 ng ml$^{-1}$ EGF, or 0.33 mM PV.

**Fluorescence lifetime imaging microscopy**. FLIM data were obtained with the Olympus FV1000 laser scanning microscope (Olympus) equipped with an external unit, PicoQuant's compact FLIM and FCS upgrade kit laser scanning microscopes (Picoquant GmbH, Berlin, Germany) using a 60×/1.35 NA Oil UPLSApo objective (Olympus). Pulsed lasers were coupled to the FV1000 through an independent port and controlled with SepiaII software (Picoquant GmbH). Detection of photons was achieved using a single photon avalanche diode (PDM Series, MPD, Picoquant GmbH) and timed using a single photon counting module (PicoHarp 300, Picoquant GmbH). Using the SymPhoTime software V5.13 (Picoquant GmbH) images were collected with an integration time of ~2 min collecting ~3.0–5.0 × 10$^6$ photons. mCitrine was excited by a 507 nm pulsed laser with an average power of 0.025 mW on the sample (LDH 507, Picoquant GmbH) and fluorescence of mCitrine was collected using a narrow-band emission filter (HQ 537/26, Chroma, Olching, Germany). FLIM data were analyzed with the global analysis code described in ref. [56] to obtain images of the mean fluorescence lifetime τ. Pixels with <50 counts were excluded from analysis. To measure the fluorescence lifetime at periphery of each cell that mostly encompasses the PM, we first derived whole cell masks, which were then eroded by 10 pixels. From the 'XOR' operation of the two masks, we obtained the 10-pixel wide peripheral masks from which the mean fluorescence lifetime τ was determined. Image processing was performed in ImageJ (http://imagej.nih.gov/ij/).

**Widefield anisotropy**. Anisotropy microscopy was performed on an Olympus IX81 inverted microscope (Olympus) equipped with a 20×/0.7 NA air objective using an Orca CCD camera (Hamamatsu Photonics, Hamamatsu City, Japan) and an incubation chamber (EMBL). mCitrine and Atto590 were excited using a MT20 illumination system. The conversion factor (electrons per digital) was determined to be 6.7 taking into account the quantum yield of the Orca CCD camera. A linear dichroic polarizer (Meadowlark optics, Frederick, CO) was implemented in the illumination path of the microscope, and two identical polarizers were placed in an external filter wheel at orientations parallel and perpendicular to the polarization of the excitation light. For each field of view two images were taken, one with the emission polarizer oriented parallel to the excitation polarizer ($I_\parallel$) and one with the emission polarizer oriented perpendicular to the excitation polarizer ($I_\perp$). The fluorescence anisotropy ($r_i$) in each pixel $i$ was calculated according to:

$$r_i = \frac{G_i I_\parallel - I_\perp}{G_i I_\parallel + 2 I_\perp}. \tag{1}$$

To determine the $G$-factor ($G_i$), parallel and perpendicular images of the fluorophore fluorescein were taken in solution. Fluorescein's anisotropy is close to zero and therefore allows calculating $G_i$ by building the ratio of the perpendicular over the parallel intensities. Fluorescence anisotropy was calculated per pixel as we expected a heterogeneous distribution of the fluorescence anisotropy within cells. To avoid bleaching only one parallel and one perpendicular image was taken, no smoothing was applied. The mean anisotropy ± SEM for binned mCitrine intensities were taken over multiple fields of view. The range of mCitrine intensity bins (1000–1800 counts) corresponded to PM localized EGFR. Every field of view contained ~5–10 cells. The CellR software supplied by the microscope manufacturer (Olympus) controlled data acquisition. Live cells were imaged in vitamin-free media at 37 °C and 5% CO$_2$ and stimulated with either 100 ng ml$^{-1}$ EGF or 0.33 mM PV.

**Quantifying CONEGI expression level**. In order to determine the expression level of CONEGI in HEK293T cells, EGF-Alexa647 (100 ng ml$^{-1}$) was administered to A431, MCF10A, and CONEGI-expressing HEK293T cells. The amount of bound EGF-Alexa647 per cell was determined by measuring the fluorescence intensity with confocal fluorescence microscopy for individual cells and integrating the intensity post-acquisition per cell. From these data, fluorescence intensity histograms were generated that represent the relative expression levels of EGFR and CONEGI in the respective cells (Supplementary Fig. 3k). In order to determine the expression level of CONEGI in HEK293T cells at which phosphorylation amplification occurs, the median of the EGF-Alexa647 fluorescence distribution was matched to that of the mCitrine fluorescence per cell distribution obtained from immunofluorescence (Fig. 3g). Assuming that MCF10A cells express 10$^5$ receptors per cell and A431 cells 10$^6$ receptors per cell allowed the estimation of CONEGI expression, which was in the range of 2–5 × 10$^5$ receptors per cell.

**Statistical analysis**. All results are expressed as mean ± SEM or in Tukey box plots. Statistical analysis was performed with GraphPad Prism, version 6.0e for Mac (GraphPad Software, La Jolla, CA, USA). Statistical significance was estimated either by paired or unpaired two-tailed $t$-tests or by two-way analysis of variance.

## Data availability

Data supporting the findings of this manuscript are available from the corresponding authors upon reasonable request.

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

## Acknowledgements

This project was partially funded by the following grants: EMBO Short-Term Fellowship (ASTF no: 122-2015) to M.B., MC_U105181009 and MC_UP_A024_1008 to J.W.C. and MRC-Nikon Case Studentship to V.B. We thank Dr. A. Sachdeva for synthesizing tetrazine-Atto590, Dr. P. Bieling and Dr. A. Krämer for critically reading the manuscript.

## Author contributions

P.I.H.B. conceived the project. M.B. designed the conformational sensor, cloned sensor constructs, and acquired and analyzed the data. C.U. and V.B. provided previously unpublished expression constructs and instruction on site-specific fluorophore labeling. M.G. analyzed anisotropy data. M.B., J.W.C., and P.I.H.B. wrote the manuscript with input from all authors.

## Additional information

**Competing interests:** The authors declare no competing interests.

