## [Peer Review File · Nature Communications]

Reviewers' comments:

Reviewer #1 (Remarks to the Author):

As asked by the editor I mainly had a look at the technical side of the imaging of the manuscript.

Overall the imaging and FRET-FLIM estimation looks fine. The used global analysis [51] is well established in the lab of the authors.

* The Fluorescence microscopy uses multiple colours, but no mentioning of bleed through to other channels. Are the channels imaged sequentially? (p24) Has a control experiment been done?

* p25. Laser power in % of the maximum are not OK. Here the power on the sample or BFP is needed.

* p26 l575: "the mean fluorescence tau"? That is the mean fluorescence lifetime tau?

* p26 l575 the cells masking in ImageJ was only based on a threshold > 50 photons? or was there additional processing. I would appreciate if they authors could describe the script or be a bit more descriptive about the procedure (as the software actually is not relevant).

* p26 the anisotropy measurements where done per pixel. What was the photon count per pixel? If the count is low than the error is rather large. Typically one would first average the intensity in the p and II channel and then do the division (Lidke et al. The role of photon statistics in fluorescence anisotropy imaging. IEEE Transactions on Image Processing, 14, 2005.) In Figure 2 and 3 the errors on the bars are very small and absolute values are also very close together indicating averages over whole cells. Please clarify.

Figure 1gh: It would be better if both y-axis had the same range to see the dependence better.

Reviewer #2 (Remarks to the Author):

The article by Baumdick et al entitled: 'A conformational sensor based on genetic code expansion reveals an autocatalytic component in EGFR activation' describes the development of a FRET sensor to investigate conformational transitions in the EGFR activation loop.

Despite several decades of investigation, the complexity of EGFR signalling has required the development of new methods that can reveal how signalling is accomplished in situ. The method employed by Baumdick et al is certainly at the frontier of what is currently possible and has the potential of providing otherwise unobtainable insights on the structural/dynamic mechanisms regulating the critical catalytic function of EGFR.

The labelling strategy involves two types of genetic modification. One is the use of genetic code expansion to substitute a single amino acid in the catalytic pocket with a designer unnatural amino acid that can be specifically labelled by the FRET acceptor. The second strategy is to incorporate mCitrine downstream of the catalytic pocket but still within the kinase domain. mCitrine will be the FRET donor.

I thoroughly enjoyed the elegant manner the FRET experiment was executed. The attention to detail here is not less than exquisite. I must say, one has grown to be accustomed to expect that from the senior author, but unfortunately this is not universal in the field. The choice of acceptor given the restrictions is the right one. The choice of donor with a single lifetime is the right one (the introduction of a double coiled-coiled to ensure rigidity to prevent donor rotations is a touch of class). The R_0 is also more than adequate considering the challenges to get fluorophore going through membranes.

No concern regarding the well-executed imaging experiments.

I also thought the authors were very candid and helpful including in the manuscript the trials of CONEGI that did not work. I believe genetic code expansion will be used more and more in the near future and the authors do a selfless service to the community reporting what has not worked.

I have got the following concerns regarding the overall experimental execution and interpretation of the results:

My main concern relates to what is actually being expressed on the cell surface by the genetic code expansion method. The efficiency of unnatural amino acid incorporation in response to the amber stop codon (UAG) in mammalian cells is commonly considered to be low. In a recent paper (Uttamapinant et al JACS 2014) the method developers state: Quote - 'One difficulty of applying genetic code expansion to protein imaging stems from the inefficiency of unnatural amino acid incorporation in mammalian cells'. The challenge is that when you use this method, unavoidably, only a fraction of receptors is transcribed incorporating the unnatural amino acid (in this publication a few %) whilst the bulk will be truncated just before the position at which the unnatural amino acid is to be inserted. In Uttamapinant et al JACS 2014 it did not matter because the amber codon was incorporated at aa 128, so the truncated receptor would not be incorporated at the plasma membrane. In Schmied et al JACS 2014, cited by the authors, an optimised pyrrolysyl-tRNA synthetase / tRNACUA and engineered eRF1 expression system was developed that provided a substantial increase in unnatural amino acid incorporation increasing read-through from 5% to approximately 30% (using luciferase instead of EGFR).

The pyrrolysyl-tRNA synthetase / tRNACUA and engineered eRF1 expression system is the one used in this work. So the key question is: How many truncated receptors find their way to the plasma membrane? These receptors are not labelled with donor, as the donor is positioned downstream of the unnatural amino acid position in the activation loop of the CONEGI. It is not inconceivable that these receptors have a partial but folded kinase domain truncated just before the position of putative unnatural amino acid incorporation. Furthermore, Y845 is upstream of the position of the successful CONEGI so the truncated receptors can therefore conceivably become phosphorylated at Y845 too. In my view, the possibility of receptors having truncated but folded kinase domains that expose receiver interfaces and can potentially engage in asymmetric kinase dimer formation needs to be excluded from the experimental design for this to become robust. On line 465 it is stated: All constructs were sequenced verified and tested for correct expression. Given the general concerns regarding the efficiency of unnatural amino acid incorporation, data stating the level of expression of truncated receptors and coping mechanisms should be provided. If the authors have found a way to bypass the challenge posed by the expression of these truncated receptors, this reviewer for one and I imagine many others in the community too would very much like to find out.

Specific questions:

The location of the mCitrine was judiciously chosen in a position that crystal structures indicate would be conformationally invariant and not block asymmetric kinase dimer formation. However, could the relative bulky mCitrine have any effect in other kinase interfaces, for example, symmetric ones? This is difficult to exclude and could potentially affect the dynamics of the system given the delicate balance between interface formation and the conformation of the catalytic pocket.

If, let's say, a percentage of receptors are truncated and therefore unlabelled but still contain Y845, how can one conclude with any certainty using FRET that Y845 phosphorylation induces a catalytically competent conformation of CONEGI monomers? If a dimer formed by a full length CONEGI construct dimerizes with a truncated non-fluorescent receptor via, for example, the receiver of the truncated receptor and the activator of the full length CONEGI this would not result in any change in the anisotropy of the mCitrine on the CONEGI, but it would still not be a

monomer. The Y845F mutation will affect both truncated and full length CONEGI, so, if there are a lot of truncated receptors, the controls do not reveal much.

Again, one would expect that truncated receptors will be able to bind EGF potentially as well as full length CONEGI ones. One would also imagine that they could form extracellularly driven EGF-induced dimers and possibly oligomers. This will affect the conclusions.

Could the conclusion that at low EGF concentration EGFR dimers can induce autocatalytic activation of monomers also potentially be explained by the presence of oligomers? If, as previously suggested (Needham et al Nat Comms 2016), larger oligomers formed at low EGF concentrations and the stoichiometry of binding is 2N:2, could amplification be distinguished from oligomerisation?

Reviewer #3 (Remarks to the Author):

The authors developed a creative and an elegant approach to monitor conformational changes in the EGFR kinase domain. Using intramolecular FRET (FLIM) they demonstrated that high density of EGFR or inhibition of protein tyrosine phosphatases by ortho-vanadate result in conformational changes that correlate with an increased tyrosine phosphorylation of the receptor, which is indicative of an autocatalytic activation of the kinase. Moreover, the authors suggest that ligand binding to and activation of a fraction of EGFRs leads to trans-activation of the fraction of ligand-free receptors through an autocatalytic mechanism. The experimental design is comprehensive, and the experiments are well controlled. The new methodology by itself deserves publication. However, there are several concerns that must be addressed.

1) Throughout the manuscript, images of Atto590 are strikingly different from corresponding images of mCitrine. The plasma membrane is not apparent in most Atto590 images, and the relative fraction of Atto590 fluorescence in the juxtannuclear compartment is significantly higher than that fraction of the mCitrine fluorescence. Such distribution pattern of Atto590 cannot be due to the interference by the cytosolic pool of Atto590. See for example Figures 1e and f, figures 2a and c, etc. Thus, it is unclear what is the nature of FLIM signals and their changes in the peripheral areas of the cell. High-resolution merged images of mCitrine and Atto590 that demonstrate clear co-localization of both signals in the plasma membrane should be presented to appreciate the effects of various experimental manipulations on FLIM.

2) The logic leading to the conclusion that there is an amplification process due to trans-autocatalytic activation of ligand-unoccupied receptors from Figure 2i (lines 289-290) is unclear. This is a critical point, and such a conclusion should be supported by the direct detection of the activity/phosphorylation of ligand-free receptors. Also, vast literature demonstrates the existence of EGFR dimers bound to one ligand (preventing binding of the second ligand due to the negative cooperativity), and this notion is not considered in the interpretation of the data in the manuscript.

3) Figure 3f. What is the nature of the substantial conformational changes observed with the Y845F mutant?

4) Figure 3g. Can the receptor density (receptor number per cell), at which the autocatalytic activation is significant, be estimated? Is such a density comparable with the EGFR expression levels in normal and cancerous cells? Such calculations are important to demonstrate the physiological importance of the autocatalytic activation.

5) Figure 4. The authors suggest that the C-terminus phosphorylation is necessary for autocatalytic activation. However, C-terminus may mediate its effects on the kinase domain

through an allosteric mechanism independently of its tyrosine phosphorylation. To prove the proposed model, Y-to-F EGFR mutants rather than receptor truncations must be utilized.

6) It appears that all lysine positions, that were used for incorporating unconventional amino acids, were previously shown to be ubiquitin-conjugation sites. Mutations of some of these lysines (Lys851?) to arginines were shown to affect receptor kinase activity. The authors may consider discussing that positions of insertion unconventional amino acids normally serve as ubiquitin-conjugation sites as well as predicted sites of the kinase domain interaction with the negatively-charged membrane lipids.

We thank the reviewers for their highly constructive comments which helped us to improve the quality of the manuscript.

Reviewer #1 (Remarks to the Author):

As asked by the editor I mainly had a look at the technical side of the imaging of the manuscript.

Overall the imaging and FRET-FLIM estimation looks fine. The used global analysis [51] is well established in the lab of the authors.

- The Fluorescence microscopy uses multiple colours, but no mentioning of bleed through to other channels. Are the channels imaged sequentially? (p24) Has a control experiment been done?

The images were acquired in sequential mode. We indeed performed control experiments and can state with confidence that we did not observe any bleed through from mCitrine in the Atto590 channel or from Atto590 in the mCitrine channel. We have added this information in the methods section (see lines 600-603; 610-611). Bleed through between Atto590 and Alexa647 can be excluded as these dyes have never been imaged together.

- p25. Laser power in % of the maximum are not OK. Here the power on the sample or BFP is needed.

The average power of the 507nm pulsed laser on the sample was measured to be 0.025mW. We have now included this information in the manuscript (see line 627).

- p26 l575: “the mean fluorescence tau”? That is the mean fluorescence lifetime tau?

This is indeed true and has been corrected in the manuscript (see line 635).

- p26 l575 the cells masking in ImageJ was only based on a threshold > 50 photons? or was there additional processing. I would appreciate if they authors could describe the script or be a bit more descriptive about the procedure (as the software actually is not relevant).

We have now better described the procedure of FLIM data analysis in the method section (see lines 631-636). We performed global analysis as described in Grecco et al., Opt Express, 2009 (now ref. 56) to obtain fluorescence lifetime images. All pixels with less than 50 donor counts were excluded from the analysis. To measure the mean fluorescence lifetime τ at the periphery of each cell that encompasses the plasma membrane we first derived whole cell masks, which were then eroded by 10 pixels. From the ‘XOR’ operation of the two masks we obtained the 10 pixel wide peripheral masks from which the mean fluorescence lifetime τ was determined.

- p26 the anisotropy measurements where done per pixel. What was the photon count per pixel? If the count is low than the error is rather large. Typically one would first average the intensity in the p and II channel and then do the division (Lidke et al. The role of photon statistics in fluorescence anisotropy imaging. IEEE Transactions on Image Processing, 14, 2005.) In Figure 2 and 3 the errors on the bars are very small

and absolute values are also very close together indicating averages over whole cells. Please clarify.

The calibration of the Orca CCD camera revealed a conversion factor (electrons per digitals) of 6.7 taking into account the quantum yield of the camera. This information was added to the method section (see lines 643-645).

It was shown for fluorescence ratio imaging (van Kempen & van Vliet, Cytometry, 2000) that averaging the ratio (i.e. the fluorescence anisotropy r) instead of the two random variables (here intensities p and II) leads to a higher value of r and an increased variance. Calculation of fluorescence anisotropy per pixel however was chosen as we expected a heterogeneous distribution of the fluorescence anisotropy within cells. To avoid bleaching only one p and II image was taken, no smoothing was applied. Fluorescence anisotropy values were consistently high for untreated cells and showed a significant drop upon stimulation with EGF due to an increase of homo-FRET. These changes are visible despite a biased estimator of r and are not masked by a larger variance of r .

The mean anisotropy \pm SEM for binned mCitrine intensities were taken over multiple fields of view. The range of mCitrine intensity bins (1000-1800 counts) corresponded to EGFR localized to the plasma membrane. Every field of view contained \sim 5-10 cells. We adapted the anisotropy analysis in the methods section (see lines 655-660) and added the total number of measured fields of view on the 3 different days of experiments to the figure legends.

Figure 1gh: It would be better if both y-axis had the same range to see the dependence better.

We now have changed the y-axis for both figures (Figure 1g,h) to the same range.

Reviewer #2 (Remarks to the Author):

The article by Baumdick et al entitled: 'A conformational sensor based on genetic code expansion reveals an autocatalytic component in EGFR activation' describes the development of a FRET sensor to investigate conformational transitions in the EGFR activation loop. Despite several decades of investigation, the complexity of EGFR signalling has required the development of new methods that can reveal how signalling is accomplished in situ. The method employed by Baumdick et al is certainly at the frontier of what is currently possible and has the potential of providing otherwise unobtainable insights on the structural/dynamic mechanisms regulating the critical catalytic function of EGFR.

The labelling strategy involves two types of genetic modification. One is the use of genetic code expansion to substitute a single amino acid in the catalytic pocket with a designer unnatural amino acid that can be specifically labelled by the FRET acceptor. The second strategy is to incorporate mCitrine downstream of the catalytic pocket but still within the kinase domain. mCitrine will be the FRET donor.

I thoroughly enjoyed the elegant manner the FRET experiment was executed. The attention to detail here is not less than exquisite. I must say, one has grown to be accustomed to expect that from the senior author, but unfortunately this is not universal in the field. The choice of acceptor given the restrictions is the right one. The choice of donor with a single lifetime is the right one (the introduction of a double coiled-coiled to ensure rigidity to prevent donor rotations is a touch of class). The R_0 is also more than adequate considering the challenges to get fluorophore going through membranes.

No concern regarding the well-executed imaging experiments.

I also thought the authors were very candid and helpful including in the manuscript the trials of CONEGI that did not work. I believe genetic code expansion will be used more and more in the near future and the authors do a selfless service to the community reporting what has not worked.

I have got the following concerns regarding the overall experimental execution and interpretation of the results:

My main concern relates to what is actually being expressed on the cell surface by the genetic code expansion method. The efficiency of unnatural amino acid incorporation in response to the amber stop codon (UAG) in mammalian cells is commonly considered to be low. In a recent paper (Uttamapinant et al JACS 2014) the method developers state: Quote - 'One difficulty of applying genetic code expansion to protein imaging stems from the inefficiency of unnatural amino acid incorporation in mammalian cells'. The challenge is that when you use this method, unavoidably, only a fraction of receptors is transcribed incorporating the unnatural amino acid (in this publication a few %) whilst the bulk will be truncated just before the position at which the unnatural amino acid is to be inserted. In Uttamapinant et al JACS 2014 it did not matter because the amber codon was incorporated at aa 128, so the truncated receptor would not be incorporated at the plasma membrane. In Schmied et al JACS 2014, cited by the authors, an optimised pyrrolysyl-tRNA synthetase / tRNACUA and engineered eRF1 expression system was developed that provided a substantial increase in unnatural amino acid incorporation increasing read-through from 5% to approximately 30% (using luciferase instead of EGFR).

The pyrrolysyl-tRNA synthetase / tRNACUA and engineered eRF1 expression system is the one used in this work. So the key question is: How many truncated receptors find their way to the plasma membrane? These receptors are not labelled with donor, as the donor is positioned downstream of the unnatural amino acid position in the activation loop of the CONEGI. It is not inconceivable that these receptors have a partial but folded kinase domain truncated just before the position of putative unnatural amino acid incorporation. Furthermore, Y845 is upstream of the position of the successful CONEGI so the truncated receptors can therefore conceivably become phosphorylated at Y845 too. In my view, the possibility of receptors having truncated but folded kinase domains that expose receiver interfaces and can potentially engage in asymmetric kinase dimer formation needs to be excluded from the experimental design for this to become robust.

On line 465 it is stated: All constructs were sequenced verified and tested for correct expression. Given the general concerns regarding the efficiency of unnatural amino acid incorporation, data stating the level of expression of truncated receptors and coping mechanisms should be provided. If the authors have found a way to bypass the challenge posed by the expression of these truncated receptors, this reviewer for one and I imagine many others in the community too would very much like to find out.

We thank this reviewer for the constructive comments and fully agree with his/her concern regarding the interaction of truncated EGFR with full-length receptor. According to the referees suggestion we have now quantified the amount of truncated EGFR in the cell to be $42 \pm 1\%$ (Supplementary figure 3c). However, an independent measure of truncated EGFR showed that $\sim 97.2 \pm 4.3\%$ of the receptors were full-length at the plasma membrane (see below). This shows that truncated EGFR did not reach the plasma membrane and therefore

our FLIM and anisotropy measurements were not affected by truncated EGFR. Please see below our more detailed response to the specific questions of the reviewer.

Specific questions:

The location of the mCitrine was judiciously chosen in a position that crystal structures indicate would be conformationally invariant and not block asymmetric kinase dimer formation. However, could the relative bulky mCitrine have any effect in other kinase interfaces, for example, symmetric ones? This is difficult to exclude and could potentially affect the dynamics of the system given the delicate balance between interface formation and the conformation of the catalytic pocket.

The known sites for the symmetric dimer interface are E991, Y992, K828, K799, R938, I942 and K946 (Zhang et al., 2006, Cell). We have added a crystal structure, where we marked the dimerization sites for the symmetric dimer and the QG insertion site of mCitrine (Supplementary figure 1a). This shows that the mCitrine insertion is not in proximity of the symmetric dimer interface. This has been amended in the text (see line 99) Moreover, the similar EGF-induced temporal phosphorylation profile and similar phosphorylation level of EGFR-QG-mCitrine as compared to C terminally tagged EGFR (EGFR-mCitrine) confirms the full functionality of EGFR-QG-mCitrine (Figure 1c).

If, let's say, a percentage of receptors are truncated and therefore unlabelled but still contain Y845, how can one conclude with any certainty using FRET that Y845 phosphorylation induces a catalytically competent conformation of CONEGI monomers? If a dimer formed by a full length CONEGI construct dimerizes with a truncated non-fluorescent receptor via, for example, the receiver of the truncated receptor and the activator of the full length CONEGI this would not result in any change in the anisotropy of the mCitrine on the CONEGI, but it would still not be a monomer. The Y845F mutation will affect both truncated and full length CONEGI, so, if there are a lot of truncated receptors, the controls do not reveal much.

We already had some data in the manuscript indicating that truncated EGFR does not reach the PM and thus does not affect our FLIM and anisotropy measurements. This is shown in Supplementary figure 3a, where we compared the anisotropy of the control EGFR-QG-mCitrine (without unnatural amino acid incorporation) to EGFR(BCNK851)-QG-mCitrine and CONEGI. The identical anisotropy changes in response to EGF for EGFR-QG-mCitrine and EGFR(BCNK851)-QG-mCitrine indicated that only full-length EGFR localizes to the PM (Supplementary Figure 3a). The smaller anisotropy change upon EGF stimulus for CONEGI is due to the ~15% decrease in mCitrine fluorescence lifetime by FRET to Atto590. We have added this reasoning in the manuscript (see lines 256-271).

However, we fully agree with the referee that the possible interference of truncated EGFR with EGFR dimerization is an essential point to be clear about, and therefore performed additional experiments. In these, we determined the amount of truncated EGFR in the cell to be $\sim 42 \pm 1\%$ using an N-terminal EGFR antibody in western blot analysis (Supplementary Figure 3c). This incorporation efficiency is higher than previously reported in the literature and could arise from proteolytic degradation of truncated EGFR. However, as our anisotropy experiments indicated that this truncated EGFR did not reach the PM, we directly measured if and how much of the truncated EGFR localizes to the PM by determining the fluorescence ratio of EGF-Alexa647 (100 ng/ml, binds to all CONEGI constructs) to mCitrine (only expressed on full length CONEGI). The identical fluorescence ratio of EGFR(BCNK851)-QG-mCitrine and full length EGFR-QG-mCitrine (Supplementary Figure 3d) confirmed that truncated CONEGI did not reach the PM. These experiments confirmed that our FLIM and

anisotropy results as well as our major conclusions were not affected by truncated EGFR. We included the new data in the manuscript as well as a description (see lines 261-266, Supplementary figure 3c,d).

Again, one would expect that truncated receptors will be able to bind EGF potentially as well as full length CONEGI ones. One would also imagine that they could form extracellularly driven EGF-induced dimers and possibly oligomers. This will affect the conclusions.

We have addressed this concern raised by the referee in the section above and showed that this is not the case. Our anisotropy data and the additionally performed EGF binding assay showed that CONEGI at the PM is full-length.

Could the conclusion that at low EGF concentration EGFR dimers can induce autocatalytic activation of monomers also potentially be explained by the presence of oligomers? If, as previously suggested (Needham et al Nat Comms 2016), larger oligomers formed at low EGF concentrations and the stoichiometry of binding is 2N:2, could amplification be distinguished from oligomerisation?

This is an interesting suggestion. However, the formation of EGFR oligomers cannot explain the difference in Y1068 phosphorylation of CONEGI as compared to its Y845F mutant. This can be concluded from the fact that at low EGF dose CONEGI potently amplifies phosphorylation in contrast to its Y845F mutant despite their similar self-association levels as measured by anisotropy (Figure 4c, Supplementary figure 4c). We have included this reasoning more extensively in the discussion of the manuscript (see lines 417-430).

Reviewer #3 (Remarks to the Author):

The authors developed a creative and an elegant approach to monitor conformational changes in the EGFR kinase domain. Using intramolecular FRET (FLIM) they demonstrated that high density of EGFR or inhibition of protein tyrosine phosphatases by ortho-vanadate result in conformational changes that correlate with an increased tyrosine phosphorylation of the receptor, which is indicative of an autocatalytic activation of the kinase. Moreover, the authors suggest that ligand binding to and activation of a fraction of EGFRs leads to trans-activation of the fraction of ligand-free receptors through an autocatalytic mechanism. The experimental design is comprehensive, and the experiments are well controlled. The new methodology by itself deserves publication. However, there are several concerns that must be addressed.

1) Throughout the manuscript, images of Atto590 are strikingly different from corresponding images of mCitrine. The plasma membrane is not apparent in most Atto590 images, and the relative fraction of Atto590 fluorescence in the juxtannuclear compartment is significantly higher than that fraction of the mCitrine fluorescence. Such distribution pattern of Atto590 cannot be due to the interference by the cytosolic pool of Atto590. See for example Figures 1e and f, figures 2a and c, etc. Thus, it is unclear what is the nature of FLIM signals and their changes in the peripheral areas of the cell. High-resolution merged images of mCitrine and Atto590 that demonstrate clear co-localization of both signals in the plasma membrane should be presented to appreciate the effects of various experimental manipulations on FLIM.

We have now included high resolution merged images to demonstrate co-localization between mCitrine and Atto590 fluorescence at the PM in Figure 1e. This revealed co-localization of

Atto590 with mCitrine at the PM, but also in intracellular compartments such as the recycling endosome through which EGFR is continuously trafficked (Supplementary figure 1f, Baumdick et al., *elife*, 2015). This co-localization is visible despite the Atto590 fluorescence in the cytosol that occurs due to unbound tet-Atto590. Together with Figure 1f this demonstrates the advantage of a donor based FRET measurement, because the control EGFR-QG-mCitrine (without BCNK) did not exhibit any decrease in mCitrine fluorescence lifetime upon Atto590 addition. The mCitrine fluorescence lifetime did only decrease in the CONEGI variants when BCNK was incorporated allowing tet-Atto590 conjugation to BCNK. This experiment verifies the specificity of the measured FRET signal.

In Figure 2c we co-expressed EGFR(BCNK851)-QG-mCitrine and PTB-mCherry, which is cytosolic in the absence of stimuli. EGF stimulation results in the recruitment of PTB-mCherry to phosphorylated EGFR.

2) The logic leading to the conclusion that there is an amplification process due to trans-autocatalytic activation of ligand-unoccupied receptors from Figure 2i (lines 289-290) is unclear. This is a critical point, and such a conclusion should be supported by the direct detection of the activity/phosphorylation of ligand-free receptors. Also, vast literature demonstrates the existence of EGFR dimers bound to one ligand (preventing binding of the second ligand due to the negative cooperativity), and this notion is not considered in the interpretation of the data in the manuscript.

It is important to note that these experiments were performed in the absence of EGF to show how autocatalytic phosphorylation depends on CONEGI density at the plasma membrane. We measured Y1068 phosphorylation and the conformational transition in separate experiments, between which we kept the microscope acquisition settings for mCitrine the same. This allowed to compare how the expression of CONEGI and its Y845F mutant relate to their conformational change and Y1068 phosphorylation. Figure 3i shows that Y1068 phosphorylation and conformational transitions of CONEGI are correlated and also depend on the CONEGI expression level. In stark contrast, there was no dependency of the conformational change and Y1068 phosphorylation on CONEGI Y845F expression. We can therefore conclude that ligand-free CONEGI monomers can adopt an active conformation upon Y845 phosphorylation showing that this site is a major factor in the amplification of EGFR phosphorylation activity in the absence of ligand (see lines 291-305). We now also determined the CONEGI expression level at which spontaneous phosphorylation occurs to fall in between endogenous EGFR expression in the non-tumorigenic epithelial MCF10A and the epidermoid carcinoma A431 cells, being closer to that of MCF10 cells (see point 4). EGFR dimers that bind one ligand will have a phosphorylation response that is linear to the fraction of liganded receptors. Oligomerization can also not bring in phosphorylation amplification what we explain in response to point 4 of referee 2 and in the discussion of the revised manuscript (see lines 417-430).

3) Figure 3f. What is the nature of the substantial conformational changes observed with the Y845F mutant?

We had shortly discussed this aspect in our manuscript (see lines 430-434). The changes in the fluorescence lifetime of the Y845F mutant showed that next to Y845 phosphorylation additional posttranslational modifications or interactions with other proteins regulated by phosphatase activity could contribute to a catalytically active conformation. The increased phosphorylation on Y1068 of CONEGI Y845F upon PV treatment as compared to basal levels (Figure 3e) also indicated that CONEGI Y845F undergoes a conformational transition

to a more active state upon PTP inhibition. However, it did not undergo the full conformational transition and reached a significantly lower Y1068 phosphorylation level as compared to CONEGI (Figure 3e,f) underlining the importance of Y845 phosphorylation in the autocatalytic amplification of EGFR phosphorylation.

The referee pointed us to the possibility of an additional post-translational modification (ubiquitination of lysines) in the activation loop that could affect this conformational transition and impact autocatalytic amplification. We added this possibility into the discussion (see lines 434-440). Please also see our response to point 6.

4) Figure 3g. Can the receptor density (receptor number per cell), at which the autocatalytic activation is significant, be estimated? Is such a density comparable with the EGFR expression levels in normal and cancerous cells? Such calculations are important to demonstrate the physiological importance of the autocatalytic activation.

We agree with the reviewer that it is essential to show at which EGFR expression levels autocatalysis occurs and therefore determined the level of CONEGI expression in an additional experiment in relation to endogenous EGFR expression in non-tumorigenic epithelial MCF10A ($\sim 1 \times 10^5$ receptors/cell) and epidermoid carcinoma A431 ($\sim 1 \times 10^6$ receptors/cell) cells. By comparing EGF-Alexa647 binding to CONEGI expressed in HEK293T cells to that of endogenous EGFR in A431 and MCF10A cells we could estimate the CONEGI expression per cell to be in the range of $\sim 2 \times 10^5$ - 5×10^5 receptors per cell. This showed that spontaneous phosphorylation occurs at all CONEGI expression levels but strongly increases at 3.8×10^5 - 4.6×10^5 receptors per cell, which falls within the lower range of endogenous EGFR expression in A431 cells. We added this new data to the manuscript (see lines 305-319; Figure 3j, Supplementary figure 3k).

5) Figure 4. The authors suggest that the C-terminus phosphorylation is necessary for autocatalytic activation. However, C-terminus may mediate its effects on the kinase domain through an allosteric mechanism independently of its tyrosine phosphorylation. To prove the proposed model, Y-to-F EGFR mutants rather than receptor truncations must be utilized.

Our truncation was based on a publication from Kovacs et al., 2015, where the authors showed that phosphorylation of Y845 vanished upon Y to A mutation of the C-terminal Y1086 (see lines 355-360). In agreement with the abovementioned paper that did not link their findings to autocatalysis, C-terminal tail deletion abolishes Y845 phosphorylation. This provided further insight in the autocatalytic activation mechanism showing that the EGFR kinase cannot directly phosphorylate Y845 on another EGFR molecule. Instead an additional tyrosine kinase that is recruited to the C-terminal tail and can phosphorylate Y845 must be involved in this mechanism.

6) It appears that all lysine positions, that were used for incorporating unconventional amino acids, were previously shown to be ubiquitin-conjugation sites. Mutations of some of these lysines (Lys851?) to arginines were shown to affect receptor kinase activity. The authors may consider discussing that positions of insertion unconventional amino acids normally serve as ubiquitin-conjugation sites as well as predicted sites of the kinase domain interaction with the negatively-charged membrane lipids.

We thank the reviewer for this valid point. Indeed K851 was shown to be a ubiquitination site (Fortian et al., Traffic, 2015), however replacement of this lysine by BCNK had no major effects on EGFR phosphorylation activity as apparent from the comparable phosphorylation levels between CONEGI and EGFR-QG-mCitrine determined by FLIM, Western blot and

immunofluorescence (Figure 2d,e,g). Together with the comment in point 3 this reviewer made us aware that ubiquitination of lysines close to the activation loop (e.g. K843) could affect EGFR activity by inducing a conformational transition in the activation loop. This could explain the conformational change in the CONEGI Y845F mutant upon PV treatment (see also point 4). However, this conformational transition was significantly smaller as compared to that of CONEGI showing the important function of Y845 phosphorylation in amplification of EGFR phosphorylation. We have now added this in the discussion as requested by the referee (see lines 434-440)

We also shortly discussed the possibility that lysine mutations could interfere with inhibitory electrostatic interactions (see lines 397-399). This indeed could be the case for mutation of K713 and K730, where we observed an increase in autonomous phosphorylation (Figure 2e). However, we did not detect an increase in autonomous phosphorylation upon K851 mutation as compared to EGFR-QG-mCitrine (Figure 2e, g) showing that the phosphorylation activity of CONEGI is not altered by BCNK incorporation.

REVIEWERS' COMMENTS:

Reviewer #1 (Remarks to the Author):

The authors fully addressed my few concerns with regard to the anisotropy calculation. They chose for an estimate with a higher variance which seemed strange methodologically, but given their experimental conditions I can understand that, good to clarify that. The minor points have been addressed to.

Reviewer #2 (Remarks to the Author):

All my concerns have been addressed satisfactorily.

Reviewer #3 (Remarks to the Author):

The authors appropriately addressed my concerns. This manuscript describes a novel elegant methodology to follow conformational changes in the EGFR molecule, which will be of great interest to the large receptor signaling community.